# Calibrating Uncertainty for Zero-Shot Adversarial CLIP

**Wenjing Lu** [1 2]  **Zerui Tao** [2]  **Yuning Qiu** [2]  **Dongping Zhang** [2 3]  **Yang Yang** [1]  **Qibin Zhao** [2]

## Abstract

CLIP delivers strong zero-shot classification but remains highly vulnerable to adversarial attacks. Prior adversarial fine-tuning work primarily matches predicted logits between clean and adversarial examples, which overlooks uncertainty calibration and may degrade the zero-shot generalization. A common expectation in reliable uncertainty estimation is that predictive uncertainty should increase as inputs become more difficult or shift away from the training distribution. However, we frequently observe the opposite in the adversarial setting: perturbations not only degrade accuracy but also suppress uncertainty, leading to severe miscalibration and over-confidence. This reveals a critical reliability gap beyond robustness. To bridge this gap, we propose an adversarial fine-tuning objective for CLIP considering both accuracy and uncertainty. By reparameterizing CLIP outputs as the concentration parameters of a Dirichlet distribution, we propose a unified representation that captures relative semantic structure and confidence magnitude. This enables holistic distribution alignment under perturbations, moving beyond single-logit anchoring and restoring calibrated uncertainty. Experiments across multiple zero-shot benchmarks demonstrate that our method significantly improves uncertainty calibration and achieves competitive adversarial robustness while preserving clean accuracy.

## 1. Introduction

Contrastive language-image pretraining (CLIP) (Radford et al., 2021) has become a widely adopted vision–language

[1]AGI Institute, School of Computer Science, Shanghai Jiao Tong University, Shanghai, China. [2]RIKEN AIP, Tokyo, Japan [3]School of Automation, Guangdong University of Technology, Guangzhou, China. Correspondence to: Yang Yang <yangyang@cs.sjtu.edu.cn>, Qibin Zhao <qibinzhao@riken.jp>.

*Proceedings of the $43^{rd}$ International Conference on Machine Learning*, Seoul, South Korea. PMLR 306, 2026. Copyright 2026 by the author(s).

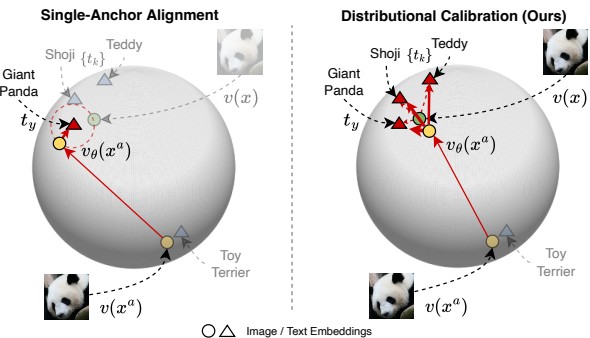

*(a)* Single-anchor alignment vs. distributional calibration.

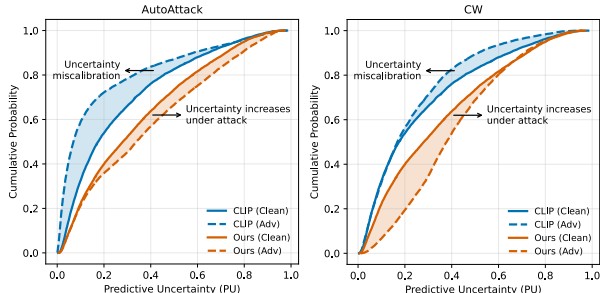

*(b)* Predictive uncertainty CDFs under AutoAttack and CW.

*Figure 1.* **Distributional alignment and uncertainty miscalibration.** (a) Prior ZSAR methods pull an adversarial image feature $v(x^a)$ toward the text prototype $t_y$ (*single-anchor*), without explicitly preserving its relative similarities to other text prototypes $\{t_k\}$. Our approach aligns predictive *distributions* to preserve inter-class semantics and evidence strength. (b) Cumulative distribution functions (CDFs) of predictive uncertainty on CIFAR-10 for clean (solid) and adversarial (dashed) samples under AutoAttack and CW ($\epsilon = 1/255$), comparing CLIP (blue) vs. Ours (orange).

model, achieving strong zero-shot recognition by comparing image features with text prompts in a shared embedding space. Its scalability (Jia et al., 2021) and adaptability through prompting or ensembling (Zhou et al., 2022; Wortsman et al., 2022) have established it as a foundation model for open-world scenarios where labeled data are scarce. Although CLIP demonstrates impressive generalization ability, it is highly vulnerable to adversarial attacks: tiny pixel-level perturbations, often imperceptible to humans, can cause confident misclassifications and severe drops in performance (Goodfellow et al., 2014; Kurakin et al., 2018; Madry et al., 2017). This contrast between strong zero-shot generalization and fragile robustness motivates the study of

adversarial reliability in vision–language models.

Recent efforts on *zero-shot adversarial robustness* (ZSAR) aim to enhance CLIP's resistance to adversarial perturbations while preserving zero-shot generalization (Mao et al., 2022; Schlarmann et al., 2024; Xing et al., 2025; Zhang et al., 2025). Formally, the task assumes that only the image encoder is adversarially fine-tuned, while the text encoder remains fixed and provides stable semantic anchors. Existing methods fine-tune the attacked encoder on labeled data to balance clean accuracy and adversarial robustness, and then evaluate transferability to unseen zero-shot datasets (Yu et al., 2024; Wang et al., 2024; Li et al., 2024b). A common strategy is to align adversarial features directly to the ground-truth text embedding, which provides strong discriminative supervision but disregards the relative geometry among neighboring classes. As illustrated in Figure 1a, the adversarial alignment is enforced only toward the ground-truth text embedding, pulling features along an unconstrained direction and disregarding the relative geometry of neighboring embeddings. However, these relations are essential as they encode inherent data ambiguity, such as semantic overlap between categories or the presence of multiple objects within a single image. Such ambiguity can be naturally interpreted as a form of predictive *uncertainty*. This single-anchor alignment provides strong discriminative supervision but neglects the underlying uncertainty structure, which can limit generalization under adversarial perturbations.

While previous methods mostly focus on aligning the predicted logits, we argue that they overlook an essential phenomenon, that is, a systematic miscalibration in CLIP's predictive uncertainty under adversarial perturbations. Figure 1b illustrates this issue on a representative benchmark, where CLIP can exhibit *lower* predictive uncertainty on adversarially perturbed inputs than on their clean counterparts. We observe the same trend across multiple datasets (see Appendix C.2 for full results), challenging the widely held expectation that uncertainty should increase with input difficulty or distributional shift (Guo et al., 2017; Hendrycks & Gimpel, 2016; Ovadia et al., 2019). This anomaly indicates that CLIP not only fails to maintain robustness but also produces spuriously confident predictions when attacked. Such behavior highlights a critical reliability gap beyond accuracy, underscoring the need to calibrate uncertainty in adversarial fine-tuning.

To address both the structural and calibration issues, we propose an Uncertainty-Calibrated Adversarial fine-Tuning (UCAT) framework for CLIP. UCAT operates by regularizing entire Dirichlet distributions rather than anchoring to a single class, thereby preserving inter-class semantic relations while calibrating the overall strength of predictive evidence. This is achieved by reparameterizing CLIP's logits as concentration parameters of a Dirichlet distribution,

yielding a unified representation for holistic alignment under perturbations. The quantitative effect of UCAT is shown in Figure 1b: compared to vanilla CLIP, our fine-tuned model achieves calibrated uncertainty levels, restoring a consistent ordering: *original CLIP w/ clean img. < fine-tuned CLIP w/ clean img. < fine-tuned CLIP w/ adversarial img.*, which faithfully reflects increasing input difficulty. The main contributions of this work can be summarized as follows:

1) **Dirichlet-based formulation of CLIP.** We reformulate CLIP's logits as concentration parameters of a Dirichlet distribution, providing a theoretically justified and closed-form approach to estimate predictive uncertainty.
2) **Uncertainty-Calibrated Adversarial fine-Tuning (UCAT).** We propose a novel uncertainty-calibrated adversarial fine-tuning method that regularizes entire Dirichlet distributions to jointly preserve inter-class relations and calibrate evidence strength.
3) **Extensive empirical validation.** Across 16 single-label benchmarks and the multi-label dataset MS-COCO, we show that our method effectively calibrates uncertainty under attack while maintaining strong clean accuracy and competitive adversarial robustness.

## 2. Related Work

**From closed-set adversarial training to open-vocabulary VLM robustness.** Classical adversarial training (AT) targets *closed-set* classification with an explicit classifier head and labeled supervision. Typical objectives jointly encourage (i) *clean discriminability* and (ii) *local robustness* around labeled samples, commonly via first-order min–max formulations (Goodfellow et al., 2014; Kurakin et al., 2018; Madry et al., 2017) or principled trade-off such as TRADES (Zhang et al., 2019b). Subsequent work improves robustness and efficiency through reweighting (Wang et al., 2019), faster inner maximization (Zhang et al., 2019a; Shafahi et al., 2019; Wong et al., 2020), weight perturbations (Wu et al., 2020), robust overfitting analyses (Rice et al., 2020), and optimization refinements (Pang et al., 2019; Addepalli et al., 2022; Cui et al., 2024).

For *open-vocabulary* vision–language models (VLMs), large-scale image–text contrastive pretraining already provides strong zero-shot recognition via image–text matching with prompts (Radford et al., 2021; Jia et al., 2021; Zhou et al., 2022). This makes the focus different from closed-set AT: the goal is not to learn a task-specific classifier for a fixed label space, but to improve robustness while keeping zero-shot transfer. Therefore, robustness is usually pursued by *adapting the pretrained model on a limited seen dataset* while *preserving the cross-modal geometry* that underpins transfer to unseen classes and datasets (Wortsman et al., 2022; Mao et al., 2022).

**Zero-shot adversarial robustness for VLMs.** Recent work improves VLM robustness under zero-shot evaluation via adversarial fine-tuning of the image encoder (Mao et al., 2022; Schlarmann et al., 2024; Wang et al., 2024; Yu et al., 2024; Li et al., 2024b; Dong et al., 2025b;a), prompt tuning (Li et al., 2024a; Shu et al., 2022; Sheng et al., 2025; Wang et al., 2025), and training-free test-time defenses (Xing et al., 2025; Tong et al., 2025; Zhang et al., 2025). Many training-time objectives adopt a *single-anchor* design that mainly enforces robustness toward the ground-truth text prototype and treats other classes as negatives (Schlarmann et al., 2024; Wang et al., 2024; Yu et al., 2024; Dong et al., 2025a), potentially under-exploiting relations among semantically related classes important for open-vocabulary evaluation.

Several methods align the clean and adversarial *softmax distributions* over the seen label set (Wang et al., 2024; Dong et al., 2025b), which mainly constrains *relative* class preferences while largely removing absolute logit-scale effects. However, calibration studies suggest that scale-related factors affect the reliability of zero-shot inference in open-vocabulary VLMs (LeVine et al., 2023; Murugesan et al., 2024), and open-set reliability often relies on confidence scores derived from *absolute* CLIP similarity values (Esmaeilpour et al., 2022). Motivated by this, we model logits as Dirichlet evidence to calibrate uncertainty and preserve both relative semantics and logit scale under attack.

**Uncertainty calibration.** Uncertainty calibration under distribution shift and adversarial perturbations has been widely studied (Guo et al., 2017; Ovadia et al., 2019). Evidential formulations provide a distributional view of predictions and can encourage high uncertainty on adversarial or OOD inputs (Malinin & Gales, 2018; 2019; Ulmer et al., 2021), while confidence-calibrated adversarial training explicitly regularizes confidence in closed-set settings (Stutz et al., 2020). Building on the discussion above that logit scale carries useful reliability signals in open-vocabulary VLMs, we adopt an evidential view: a Dirichlet prediction separates the *relative* class structure (the shape of the Dirichlet, often associated with aleatoric uncertainty) from the *overall* evidence strength (often associated with epistemic uncertainty) (Sensoy et al., 2018; Malinin & Gales, 2018; Ulmer et al., 2021; Ma et al., 2025). We leverage this decomposition to formulate a scale-aware alignment objective that couples distributional semantics with evidence strength, yielding calibrated confidence under attack.

## 3. Preliminary

### 3.1. Zero-Shot Vision-Language Models

**Contrastive pretraining objective.** Contrastive pretraining underlies large-scale vision–language models such as CLIP (Radford et al., 2021). Let $f_\theta : \mathcal{X}_{\text{img}} \rightarrow$ $\mathbb{R}^d$, $g_\phi : \mathcal{X}_{\text{txt}} \rightarrow \mathbb{R}^d$ denote the image and text encoders, where $d$ is the dimension of the embedding space. For an image–text pair $(x_i^{\text{img}}, x_i^{\text{txt}})$, the embeddings are normalized onto the unit hypersphere $\mathbb{S}^{d-1}$: $v_i = f_\theta(x_i^{\text{img}})/\|f_\theta(x_i^{\text{img}})\|_2$, $t_i = g_\phi(x_i^{\text{txt}})/\|g_\phi(x_i^{\text{txt}})\|_2$. The similarity between image $i$ and text $j$ can be expressed in two directional forms: $\ell_{ij}^{v \rightarrow t} = \langle v_i, t_j \rangle/\tau$, $\ell_{ij}^{t \rightarrow v} = \langle t_i, v_j \rangle/\tau$, where $\tau > 0$ is a learnable temperature parameter. Given a batch of $N$ aligned pairs, the symmetric InfoNCE objective is

$$\mathcal{L}_{\text{InfoNCE}} = -\frac{1}{2N} \sum_{i=1}^{N} \sum_{d \in \{v \rightarrow t,\, t \rightarrow v\}} \log \frac{\exp(\ell_{ii}^d)}{\sum_{j=1}^{N} \exp(\ell_{ij}^d)}. \tag{1}$$

**Zero-shot Classification.** Under contrastive pretraining, CLIP performs zero-shot classification by matching images to text prompts in a shared embedding space via the image-to-text similarity $\ell^{v \rightarrow t}$ (Jia et al., 2021; Yao et al., 2021; Zhai et al., 2022; Zhou et al., 2022). Each class label $c_k$ ($k = 1, \ldots, C$, where $C$ is the number of candidate classes) is converted into a natural-language prompt (e.g., "This is a photo of a dog"), which is encoded and normalized to yield a class prototype $t_k \in \mathbb{S}^{d-1}$. For a test image $x$, the normalized embedding is $v(x) = f_\theta(x)/\|f_\theta(x)\|_2$, and the logit for class $c_k$ is $\ell_k^{v \rightarrow t}(x) = \langle v(x), t_k \rangle/\tau$. The predictive distribution over classes is obtained via the softmax

$$p^{\text{CLIP}}(y = k \mid x) = \frac{\exp(\ell_k^{v \rightarrow t}(x))}{\sum_{j=1}^{C} \exp(\ell_j^{v \rightarrow t}(x))}. \tag{2}$$

This formulation enables recognition of categories unseen during training, relying solely on the shared image–text embedding space.

### 3.2. Adversarial Attacks

Adversarial attacks perturb inputs with small, often imperceptible changes to mislead a model. Given an image $x$ with label $y$, an adversarial example is constructed as $x^a = x + \delta, \|\delta\|_q \leq \epsilon$, where $\epsilon$ bounds the perturbation magnitude under $\ell_q$-norm. A canonical method is *Projected Gradient Descent* (PGD, Madry et al., 2017). For the common $\ell_\infty$ threat model, it iteratively updates

$$x_{t+1}^a = \Pi_{B_\epsilon^\infty(x)}\Big(x_t^a + \alpha \, \text{sign}\big(\nabla_x \mathcal{L}(F_\varphi(x_t^a), y)\big)\Big), \tag{3}$$

where $B_\epsilon^\infty(x) = \{x' : \|x' - x\|_\infty \leq \epsilon\}$, $t$ is the iteration index, $\alpha$ is the step size, $F_\varphi$ is the target model, and $\Pi_{B_\epsilon^\infty(x)}$ projects back to the feasible set. Intuitively, PGD takes a step that increases the loss and then clips the perturbed image to stay within the allowed budget.

### 3.3. Uncertainty Estimation via Evidence

**Dirichlet Parameterization with Evidence.** In evidential deep learning (EDL), predictive uncertainty is mod-

eled explicitly by placing a *Dirichlet distribution* over class probabilities rather than predicting a single categorical distribution (Sensoy et al., 2018; Malinin & Gales, 2018; Ulmer et al., 2021). For a $C$-class problem, the network outputs non-negative concentration parameters $\alpha = (\alpha_1, \ldots, \alpha_C) \in \mathbb{R}^C_+$, typically expressed as $\alpha_k = e_k + 1, e_k \geq 0$, where $e_k$ denotes the evidence assigned to class $k$. In the original EDL formulation, this ensures $\alpha_k \geq 1$ so that zero evidence corresponds to a uniform prior. The induced Dirichlet distribution is

$$\mathrm{Dir}(\pi; \alpha) = \frac{1}{B(\alpha)} \prod_{k=1}^C \pi_k^{\alpha_k - 1}, \quad B(\alpha) = \frac{\prod_{k=1}^C \Gamma(\alpha_k)}{\Gamma(\alpha_0)},$$

(4)

where $\pi = (\pi_1, \ldots, \pi_C)$ is a probability on the $(C-1)$-simplex and $B(\alpha)$ is the polynomial Beta function. Importantly, $\alpha_0 = \sum_{k=1}^C \alpha_k$ quantifies the total evidence and serves as the precision of the distribution.

The non-negativity of $\alpha$ is typically enforced by activation functions such as ReLU, Softplus, or exponential mapping used in prior works (Yoon & Kim, 2024; Malinin & Gales, 2019). In particular, under the exponential parameterization with unconstrained logits $z(x) \in \mathbb{R}^C$ and $\alpha_k(x) = \exp(z_k(x))$, the predictive categorical distribution is obtained as the expectation under the Dirichlet:

$$\begin{aligned} p(y = k \mid x) &:= \mathbb{E}_{\pi \sim \mathrm{Dir}(\alpha(x))}[\pi_k] \\ &= \frac{\alpha_k(x)}{\alpha_0(x)} \stackrel{\alpha_k = \exp(z_k)}{=} \frac{\exp(z_k(x))}{\sum_{j=1}^C \exp(z_j(x))}. \end{aligned}$$

(5)

**Closed-Form Uncertainty Decomposition.** The Dirichlet parameterization not only provides a probability distribution but also admits a closed-form decomposition of predictive uncertainty into two complementary components, aleatoric and epistemic (Kiureghian & Ditlevsen, 2009; Kendall & Gal, 2017; Hüllermeier & Waegeman, 2021).

*Aleatoric uncertainty (AU)* captures ambiguity inherent in the data. In vision–language models, this may arise from factors such as semantic overlap between classes (e.g., "wolf" vs. "dog") or noisy image–text pairs where multiple labels are plausible (Ulmer et al., 2021; Ma et al., 2025; Ji et al., 2023). Formally, AU reflects how probability mass is distributed across classes and is quantified by the expected Shannon entropy of the categorical distribution under the Dirichlet:

$$\begin{aligned} \mathrm{AU}(x) &= \mathbb{E}_{\pi \sim \mathrm{Dir}(\alpha)}[H(\pi)] \\ &= -\sum_{k=1}^C \frac{\alpha_k}{\alpha_0} \Big( \psi(\alpha_k + 1) - \psi(\alpha_0 + 1) \Big), \end{aligned}$$

(6)

where $\psi(\cdot)$ denotes the digamma function.

*Epistemic uncertainty (EU)* arises from limited evidence

or distributional shift (Hendrycks & Gimpel, 2016; Sensoy et al., 2018). It reflects the overall reliability of the prediction: when the total evidence $\alpha_0$ is small, the model should be considered untrustworthy. Following prior work (Charpentier et al., 2020; Ulmer et al., 2021; Ma et al., 2025), a widely adopted closed-form proxy is

$$\mathrm{EU}(x) = \frac{C}{\alpha_0 + C},$$

(7)

which increases as $\alpha_0$ decreases.

In summary, AU reflects ambiguity in the predictive distribution across classes, while EU captures uncertainty from insufficient evidence or distributional shift. Both can be computed directly from the Dirichlet parameters, enabling efficient uncertainty estimation in a single forward pass.

## 4. Dirichlet Reformulation of CLIP

Comparing CLIP's zero-shot probability in Equation 2 with the Dirichlet expectation in Equation 5 reveals a structural correspondence: both are softmax operations over a set of logits. This motivates a *non-trivial* identification that reinterprets CLIP logits as *evidence* governing a Dirichlet distribution (Definition 4.1). This identification is non-trivial for three reasons: (i) it satisfies the validity of Dirichlet evidence with tight bounds and strict monotonicity (Lemma 4.3); (ii) it exactly recovers CLIP's predictive rule exactly under a specific calibration (Lemma 4.5); and (iii) preserves logit order while exposing a tunable temperature for calibration (Corollary 4.7).

**Definition 4.1** (Concentration Parameter). Let $v(x), t_k \in \mathbb{S}^{d-1}$ be unit-normalized image/text embeddings and $\ell_k^{v \to t}(x) = \langle v(x), t_k \rangle / \tau$ the CLIP logit with temperature $\tau > 0$. We define Dirichlet concentration parameters by

$$\alpha_k(x) = \exp\big(h(\ell_k^{v \to t}(x))\big), \qquad h(\ell) = \frac{\tau \ell + 1}{\tau'}, \quad (8)$$

where $\tau' > 0$ is a calibration coefficient.

*Remark* 4.2 (Construction rationale). Since $\tau \, \ell_k^{v \to t}(x) = \langle v(x), t_k \rangle \in [-1, 1]$, we shift the cosine similarity by $+1$ so that its range becomes $[0, 2]$. A calibration coefficient $\tau' > 0$ is introduced to rescale. Applying the exponential guarantees positivity while preserving logit order and remaining compatible with softmax geometry.

**Lemma 4.3** (Validity of Dirichlet Evidence). *Under Definition 4.1, for all $k$:*

1. $\alpha_k(x) \geq 1$ *and* $\alpha_k(x) \in [1, \exp(2/\tau')]$;
2. $\alpha = \exp(h(\ell))$ *is strictly increasing.*

*Remark* 4.4 ($\alpha_k \geq 1$ in EDL). As introduced in Section 3.3, the classical EDL formulation enforces $\alpha_k \geq 1$ by parameterizing $\alpha_k = e_k + 1$ with non-negative evidence (Sensoy

et al., 2018; 2020). We adopt the same restriction for two reasons: (i) digamma- and trigamma-based uncertainty measures become unstable as $\alpha_k$ approaches 0 (Minka, 2000), and (ii) Dirichlet distributions with $\alpha_k < 1$ produce corner-seeking samples (Telgarsky, 2013), concentrating on a few classes even under weak evidence. This violates the common principle that uncertainty should grow as inputs become harder or deviate from the training distribution. Accordingly, our reformulation guarantees $\alpha_k \geq 1$; all subsequent analysis and experiments are under this regime. Proof is provided in Appendix B.1.

**Lemma 4.5** (Exact Equivalence at $\tau = \tau'$). *Let $s = \tau/\tau'$. If $s = 1$ (equivalently $\tau' = \tau$), the Dirichlet expectation equals to CLIP's softmax:*

$$p_k^{\text{Dir}}(x) = \frac{\alpha_k}{\sum_j \alpha_j} = \frac{\exp(h(\ell_k))}{\sum_j \exp(h(\ell_j))} = p_k^{\text{CLIP}}(x). \tag{9}$$

*Remark* 4.6 (Significance of exact equivalence). Lemma 4.5 shows that when $\tau' = \tau$, the Dirichlet expectation coincides exactly with CLIP's softmax prediction. This equivalence is not incidental: it demonstrates that CLIP's original training loss in Equation 1 admits a Dirichlet-consistent reinterpretation. Hence, our reformulation is not an ad hoc construction but a faithful probabilistic interpretation of CLIP's logits. A complete proof is provided in Appendix B.2.

**Corollary 4.7** (General form and invariances). *For arbitrary $\tau' > 0$, $s = \tau/\tau' > 0$, $p^{\text{Dir}}(x) = \text{softmax}\big(s\,\ell(x)\big)$. Hence*

$$\arg\max_k p_k^{\text{Dir}}(x) = \arg\max_k p_k^{\text{CLIP}}(x), \tag{10}$$

*while the entropy of the distribution can be smoothly tuned by $s$: larger $s$ yields sharper predictions, smaller $s$ yields flatter ones.*

*Remark* 4.8 (Connection to uniformity–tolerance in contrastive learning). In contrastive learning, the temperature regulates the separation strength among negatives. A *smaller* softmax temperature (larger $s$) encourages *uniformity* on the hypersphere by enforcing stronger separation, while a *larger* temperature (smaller $s$) increases *tolerance* to near-semantic neighbors (Wang & Isola, 2020; Wang & Liu, 2021). We set $\tau' = 0.07$, a canonical temperature in contrastive learning (He et al., 2020; Radford et al., 2021; Jia et al., 2021). As shown in Appendix B.3, this implies $s < 1$ and thus softer predictions with increased tolerance to semantically related negatives. Sensitivity analysis over $\tau'$ is reported in Appendix D.4. This choice maintains the relative similarity structure induced by CLIP and is used throughout our adversarial fine-tuning experiments to control the tolerance level.

**Implications of the reformulation.** This reformulation establishes a principled mapping from CLIP logits to Dirich-

let evidence. Specifically, it enables: (i) **closed-form uncertainty decomposition** into AU/EU without sampling (Section 3.3); (ii) **principled calibration** via $\tau'$, which adjusts predictive sharpness while preserving the argmax (Corollary 4.7) and thus exposes a controllable uniformity–tolerance trade-off; and (iii) **semantic fidelity**, recovering CLIP's predictive rule in the exact-equivalence case (Lemma 4.5) while retaining both relative geometry and absolute evidence strength.

These properties directly motivate the adversarial fine-tuning objectives introduced in the next section.

## 5. Uncertainty Calibration Adversarial Fine-tuning Objective

To mitigate the misaligned semantics and unreliable confidence introduced by adversarial perturbations, we propose an *Uncertainty Calibration Adversarial fine-Tuning (UCAT)* objective. The key insight builds on our reformulation: mapping CLIP logits to Dirichlet evidence yields closed-form uncertainty decomposition with principled calibration, while retaining fidelity to the semantic geometry of the embedding space. UCAT exploits this property by aligning the Dirichlet distributions of adversarial and clean samples, correcting distributional shift while simultaneously preserving *semantic relations* and *calibrated confidence*.

As illustrated in Figure 2, our method adopts a CLIP-based adversarial fine-tuning pipeline with a frozen text encoder and a trainable image encoder. Clean samples $x$ and their adversarial counterparts $x^a$ (generated via $\ell_\infty$-PGD (Madry et al., 2017)) are encoded into the joint embedding space, and their logits are reformulated as Dirichlet parameters, denoted $\alpha$ and $\alpha_{\text{adv}}$. The clean distribution $\text{Dir}(\alpha)$ captures the generalized semantics from pre-training, whereas $\text{Dir}(\alpha_{\text{adv}})$ may shift toward distorted or overconfident states. To correct this mismatch, we introduce an *uncertainty calibration regularization* objective, defined as the reverse KL divergence between the two distributions:

$$\mathcal{L}_{\text{ucr}} = \text{KL}(\text{Dir}(\alpha_{\text{adv}}) \,\|\, \text{Dir}(\alpha)). \tag{11}$$

Our choice of the reverse KL direction follows naturally from the distribution alignment objective and significantly impacts optimization. Unlike the forward KL, which covers modes and flattens evidence, the reverse KL is mode-seeking. It preserves both relative class structure and absolute evidence strength by allowing low evidence on irrelevant classes (Malinin & Gales, 2019). Since both AU and EU are closed-form functions of Dirichlet parameters (Sec. 3.3), minimizing $\mathcal{L}_{\text{ucr}}$ aligns adversarial predictions with their clean counterparts in terms of *inter-class relations* (AU) and *evidence magnitude* (EU). This alignment prevents collapse into spuriously confident errors, enhancing uncertainty estimation and zero-shot adversarial robustness.

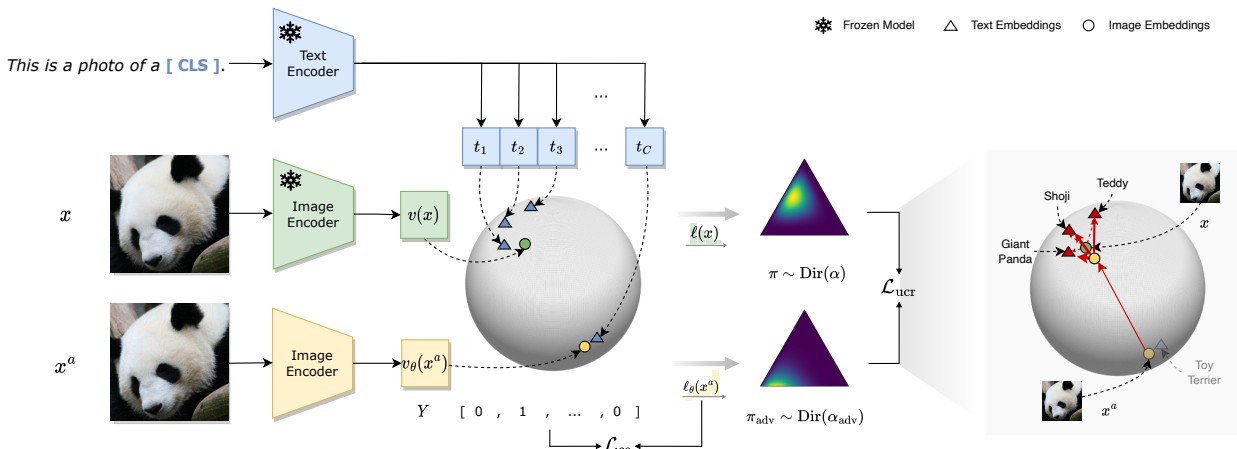

*Figure 2.* **Overview of our uncertainty calibration adversarial fine-tuning framework.** Clean and adversarial images are encoded by CLIP's image encoder, while text prompts are processed by the frozen text encoder. Our training objective combines the text-guided contrastive loss $\mathcal{L}_{ce}$ with an uncertainty calibration regularization term $\mathcal{L}_{ucr}$ that aligns adversarial Dirichlet distributions with the corresponding clean Dirichlet distributions, aiming to better preserve semantic relations and calibrate evidence strength.

*Table 1.* **Zero-shot adversarial robustness on multi-label dataset MS-COCO (Lin et al., 2014).** All ZSAR models are adversarially trained on TinyImageNet with the FARE (Schlarmann et al., 2024) 10-step PGD setting ($\epsilon = 1/255$), and evaluated under CW-100 at radii $\epsilon \in \{1/255, 2/255, 4/255\}$ plus clean. We report mean Average Precision (mAP), Precision (P), Recall (R), and F1-score (F1) at top-3 predictions. $H(\text{F1@3})$ denotes the harmonic mean of clean and adversarial F1@3. Best and second-best are in **bold** and underline.

| | Clean | | | | $\epsilon = 1/255$ | | | | | $\epsilon = 2/255$ | | | | | $\epsilon = 4/255$ | | | | |
| Methods | mAP | P@3 | R@3 | F1@3 | mAP | P@3 | R@3 | F1@3 | $H_{(\text{F1@3})}$ | mAP | P@3 | R@3 | F1@3 | $H_{(\text{F1@3})}$ | mAP | P@3 | R@3 | F1@3 | $H_{(\text{F1@3})}$ |
|---|---|---|---|---|---|---|---|---|---|---|---|---|---|---|---|---|---|---|---|
| CLIP (Radford et al., 2021) | 51.96 | 45.33 | 46.49 | 45.90 | 17.72 | 25.21 | 25.85 | 25.52 | 32.80 | 6.67 | 10.07 | 10.32 | 10.19 | 16.68 | 3.77 | 3.84 | 3.94 | 3.89 | 7.17 |
| TeCoA (Mao et al., 2022) | 46.34 | 33.73 | 34.57 | 34.14 | 37.32 | 30.23 | 30.99 | 30.60 | 32.27 | 27.94 | 24.85 | 25.48 | 25.16 | 28.97 | **17.63** | 19.67 | 20.17 | 19.91 | 25.15 |
| FARE (Schlarmann et al., 2024) | **49.21** | 43.83 | 44.95 | 44.37 | 29.18 | 33.45 | 34.30 | 33.86 | 38.41 | 16.18 | 22.44 | 23.02 | 22.72 | 30.05 | 8.12 | 13.21 | 13.55 | 13.38 | 20.56 |
| PMG-AFT (Wang et al., 2024) | 48.94 | 41.77 | 42.83 | 42.29 | 29.75 | 32.32 | 33.15 | 32.72 | 36.89 | 17.16 | 23.37 | 23.98 | 23.67 | 30.35 | 8.78 | 13.45 | 13.80 | 13.62 | 20.60 |
| TGA-ZSR (Yu et al., 2024) | 48.61 | 37.19 | 38.13 | 37.65 | **38.23** | 32.95 | 33.79 | 33.36 | 35.38 | 29.03 | 27.70 | 28.40 | 28.04 | 32.14 | 15.18 | 18.64 | 19.11 | 18.87 | 25.14 |
| Comp-TGA (Yu et al., 2026) | 48.14 | 38.11 | 39.07 | 38.58 | 37.57 | 33.05 | 33.88 | 33.45 | 35.83 | 27.23 | 28.16 | 28.87 | 28.51 | 32.79 | 13.61 | 18.08 | 18.54 | 18.30 | 24.82 |
| UCAT (Ours) | 47.14 | 41.55 | 42.62 | 42.07 | 37.60 | 36.58 | 37.52 | 37.04 | 39.40 | 29.33 | 31.39 | 32.20 | 31.78 | 36.21 | 14.61 | 21.33 | 21.86 | 21.59 | 28.54 |

Complementarily, the text-guided cross-entropy loss

$$\mathcal{L}_{ce} = -\log \frac{\exp\left(\langle v(x^a), t_y \rangle / \tau\right)}{\sum_{j=1}^{C} \exp\left(\langle v(x^a), t_j \rangle / \tau\right)}, \qquad (12)$$

anchors adversarial embeddings to the ground-truth prototype $t_y$, providing discriminative supervision that stabilizes training and improves accuracy. The final objective combines both components:

$$\mathcal{L} = \mathcal{L}_{ce} + \lambda\, \mathcal{L}_{ucr}, \qquad (13)$$

where $\lambda$ balances discriminative alignment and uncertainty calibration. This joint objective combines discriminative supervision via the cross-entropy loss with calibrated uncertainty through distributional alignment, leading to stronger zero-shot adversarial robustness.

## 6. Experiments

**Experimental Setup.** We adopt CLIP-B/32 (Radford et al., 2021) as the backbone and follow TeCoA's training protocol (Mao et al., 2022), comparing zero-shot adversarial robustness against five baselines: CLIP (Radford et al., 2021), TeCoA, FARE (Schlarmann et al., 2024), PMG-AFT (Wang

et al., 2024), and TGA-ZSR (Yu et al., 2024). Training and evaluation are conducted under $\ell_\infty$ PGD regimes, including a light setting (2-step, $\epsilon = 1/255$) following TeCoA and a stronger setting (10-step, $\epsilon = 2/255$) following FARE. Robustness is further assessed using 100-step PGD (Madry et al., 2017), CW (Carlini & Wagner, 2017), and AutoAttack (Croce & Hein, 2020). We set $\lambda = 10^5/\beta$ with $\beta = 2/e^{\tau'}$, and fix $\tau' = 0.07$ following standard contrastive learning practices (Wu et al., 2018; He et al., 2020; Radford et al., 2021; Yeh et al., 2022). Full implementation details and datasets are provided in the Appendix A. Code is available at https://github.com/VivienLu/UCAT.

### 6.1. Robustness under Multi-Label Ambiguity

To assess robustness under multi-label ambiguity, we perform zero-shot evaluation on the multi-label MS-COCO (Lin et al., 2014) dataset (Table 1). All models are fine-tuned on single-label TinyImageNet using PGD and tested directly on COCO under CW attacks that perturb multiple labels simultaneously. Across three attack strengths, our method consistently achieves the best top-3 precision, recall, F1, and harmonic mean of clean and adversarial F1@3, indicating a superior accuracy–robustness trade-off for the top-ranked predictions.

*Table 2.* **Zero-shot adversarial robustness across 16 single-label datasets.** All methods are fine-tuned on TinyImageNet following TGA-ZSR (Yu et al., 2024), adversarial training uses 2-step PGD (Madry et al., 2017) with $\epsilon = 1/255$. *Average* is the mean across datasets. $H$ is the harmonic mean between Clean and the corresponding robust score. Best and second-best are in **bold** and underline.

| Methods | TinyImageNet | CIFAR-10 | CIFAR-100 | STL10 | SUN397 | Food101 | Oxfordpets | Flowers102 | DTD | EuroSAT | FGVC Aircraft | ImageNet | Caltech101 | Caltech256 | StanfordCars | PCAM | Average | H |
|---|---|---|---|---|---|---|---|---|---|---|---|---|---|---|---|---|---|---|
| **Clean** | | | | | | | | | | | | | | | | | | |
| CLIP (Radford et al., 2021) | 57.96 | 88.03 | 60.45 | 97.03 | 57.26 | 83.89 | 87.41 | 65.49 | 40.64 | 42.66 | 20.16 | 59.15 | 85.32 | 81.73 | 52.02 | 52.08 | 64.45 | |
| TeCoA (Mao et al., 2022) | 71.24 | 67.56 | 38.26 | 85.89 | 36.01 | 28.23 | 61.30 | 32.04 | 24.95 | 16.13 | 5.19 | 32.89 | 72.16 | 59.00 | 20.28 | 50.11 | 43.83 | |
| FARE (Schlarmann et al., 2024) | 41.86 | 79.81 | 48.27 | **94.24** | 46.15 | **58.90** | **80.98** | 47.63 | 23.09 | **24.19** | **15.63** | 42.93 | 78.22 | 72.05 | **43.96** | 50.02 | 53.00 | |
| PMG-AFT (Wang et al., 2024) | 48.60 | 74.73 | 43.59 | 90.41 | **51.70** | 56.52 | 79.40 | **48.43** | **32.45** | 21.76 | 11.79 | **46.74** | **82.49** | **73.59** | 41.21 | **56.13** | 53.72 | |
| TGA-ZSR (Yu et al., 2024) | **76.60** | 79.18 | 47.37 | 90.65 | 43.10 | 38.90 | 68.44 | 39.81 | 25.69 | 19.70 | 8.82 | 39.27 | 76.42 | 66.31 | 28.44 | 49.92 | 49.91 | |
| Comp-TGA (Yu et al., 2026) | 75.00 | **81.91** | 50.67 | 90.84 | 46.45 | 44.28 | 72.04 | 41.41 | 28.35 | 23.50 | 9.12 | 42.23 | 78.96 | 69.25 | 29.55 | 49.88 | 52.09 | |
| UCAT (Ours) | 74.46 | 81.81 | **54.45** | 91.88 | 41.06 | 53.58 | 74.16 | 47.57 | 31.92 | 19.29 | 10.95 | 43.20 | 82.39 | 71.53 | 37.32 | 51.20 | **54.17** | |
| **PGD** | | | | | | | | | | | | | | | | | | |
| CLIP (Radford et al., 2021) | 0.19 | 9.57 | 3.07 | 23.64 | 0.62 | 0.34 | 0.64 | 1.62 | 2.22 | 0.00 | 0.00 | 0.48 | 5.65 | 7.19 | 0.02 | 0.06 | 3.46 | 6.56 |
| TeCoA (Mao et al., 2022) | 50.96 | 39.33 | 21.64 | 69.78 | 20.07 | 13.50 | 37.80 | 19.17 | 18.30 | **11.88** | 2.16 | 18.47 | 56.00 | 42.38 | 9.33 | 46.92 | 29.86 | 35.52 |
| FARE (Schlarmann et al., 2024) | 3.78 | 7.83 | 2.80 | 48.18 | 5.66 | 2.45 | 6.52 | 5.75 | 0.08 | 0.54 | | 5.20 | 33.21 | 20.70 | 2.31 | 48.97 | 12.81 | 20.63 |
| PMG-AFT (Wang et al., 2024) | 19.18 | **51.39** | 27.23 | 72.63 | 20.05 | 16.88 | 44.59 | 26.43 | **20.05** | 11.49 | 3.21 | 18.09 | 61.13 | 43.46 | 14.80 | **55.52** | 31.63 | 39.82 |
| TGA-ZSR (Yu et al., 2024) | **50.68** | 42.16 | 22.82 | 72.18 | 21.57 | 16.53 | 39.96 | 22.44 | 17.82 | 11.75 | 2.88 | 20.39 | 58.05 | 46.18 | 11.40 | 48.05 | 31.55 | 38.66 |
| Comp-TGA (Yu et al., 2026) | 49.90 | 40.44 | 22.43 | 72.56 | **21.61** | 16.81 | 41.56 | 23.04 | 18.88 | 11.66 | 2.70 | 20.07 | 58.80 | 46.42 | 10.50 | 44.98 | 31.40 | 39.18 |
| UCAT (Ours) | 47.56 | 43.81 | 25.16 | 73.83 | 20.44 | 22.86 | 45.11 | 26.79 | 19.47 | 2.99 | 3.45 | 22.22 | 65.32 | 50.47 | 15.30 | 30.37 | 32.20 | 40.39 |
| **CW** | | | | | | | | | | | | | | | | | | |
| CLIP (Radford et al., 2021) | 0.14 | 9.91 | 3.34 | 26.01 | 1.16 | 0.51 | 0.87 | 2.03 | 2.55 | 0.01 | 0.00 | 1.10 | 6.82 | 8.17 | 2.32 | 0.04 | 4.06 | 7.64 |
| TeCoA (Mao et al., 2022) | 50.16 | 38.62 | 20.76 | 69.55 | 18.84 | 12.46 | 37.37 | 18.12 | 17.23 | **11.63** | 2.10 | 17.70 | 55.62 | 41.70 | 9.23 | 46.88 | 29.25 | 35.08 |
| FARE (Schlarmann et al., 2024) | 4.10 | 4.12 | 2.96 | 43.35 | 6.07 | 3.17 | 15.15 | 5.66 | 4.52 | 0.12 | 1.11 | 5.34 | 32.50 | 20.85 | 4.38 | 48.86 | 12.64 | 20.41 |
| PMG-AFT (Wang et al., 2024) | 13.16 | 42.10 | 21.31 | 65.69 | 13.12 | 11.43 | 28.05 | 17.53 | 12.55 | 8.51 | 0.99 | 11.72 | 52.84 | 35.68 | 7.06 | 14.26 | 22.25 | 31.47 |
| TGA-ZSR (Yu et al., 2024) | **50.80** | 42.24 | 22.64 | 71.99 | 20.83 | 16.03 | 40.20 | 21.52 | 16.97 | 11.56 | 2.85 | 20.01 | 57.72 | 45.84 | 11.23 | **48.03** | 31.28 | 38.46 |
| Comp-TGA (Yu et al., 2026) | 49.82 | 40.40 | 22.43 | 72.33 | **21.13** | | 42.25 | 22.23 | 17.55 | 11.41 | 2.64 | 19.94 | 58.66 | 46.13 | 11.09 | 45.01 | 31.16 | 38.99 |
| UCAT (Ours) | 47.08 | **43.30** | 23.92 | 73.55 | 19.20 | 21.68 | 45.38 | 24.95 | 17.87 | 2.41 | **3.21** | 21.14 | 64.63 | 49.54 | 14.75 | 29.89 | 31.41 | 39.76 |
| **Auto Attack** | | | | | | | | | | | | | | | | | | |
| CLIP (Radford et al., 2021) | 0.00 | 2.54 | 1.11 | 3.18 | 0.05 | 0.03 | 0.03 | 0.02 | 0.19 | 0.17 | 0.23 | 0.04 | 0.10 | 0.26 | 0.07 | 0.12 | 0.51 | 1.01 |
| TeCoA (Mao et al., 2022) | **49.44** | 37.87 | 20.45 | 69.31 | 17.41 | 12.19 | 36.58 | 17.81 | 17.29 | 11.42 | 1.86 | 17.19 | 54.95 | 41.19 | 8.16 | 46.79 | 28.74 | 34.72 |
| FARE (Schlarmann et al., 2024) | 0.12 | 0.03 | 0.21 | 10.18 | 0.84 | 0.19 | 0.93 | 0.60 | 1.92 | 0.07 | 0.06 | 0.86 | 10.26 | 5.59 | 0.21 | 5.15 | 2.33 | 4.45 |
| PMG-AFT (Wang et al., 2024) | 8.22 | 41.86 | 21.18 | 65.45 | 7.95 | 7.34 | 18.94 | 12.59 | 3.13 | 7.17 | 0.51 | 7.90 | 44.91 | 28.29 | 3.22 | 7.41 | 17.88 | 26.83 |
| TGA-ZSR (Yu et al., 2024) | 49.26 | 40.92 | 21.75 | 71.55 | **19.88** | 15.32 | 38.84 | 20.98 | 17.02 | 11.26 | 2.34 | 19.12 | 57.11 | 45.16 | 9.87 | **48.00** | 30.52 | 37.88 |
| Comp-TGA (Yu et al., 2026) | 39.84 | 30.11 | 17.23 | 65.48 | 15.70 | 12.50 | 34.10 | 18.21 | 14.84 | 10.82 | 1.74 | 15.87 | 52.61 | 40.69 | 7.38 | 42.81 | 26.24 | 34.90 |
| UCAT (Ours) | 45.80 | 42.32 | 23.03 | 73.15 | 18.26 | 20.52 | 44.02 | 24.54 | 18.14 | 2.26 | **2.61** | 20.15 | 63.73 | 48.66 | 12.60 | 29.51 | 30.58 | 39.09 |

Compared with single-anchor approaches that align adversarial visual features to single-category text features or original visual features, our method aligns clean and adversarial Dirichlet distributions in the evidence (similarity) space. This distributional alignment preserves both relative semantic ordering among categories and absolute evidence strength, without enforcing softmax-based cross-class normalization, thereby stabilizing semantically relevant labels near the head of the ranking under multi-label ambiguity. Consequently, improvements are more pronounced on metrics emphasizing top-$k$ reliability, while gains in mAP are more limited, as mAP evaluates fine-grained global ordering across all labels, including low-confidence and long-tail categories. Overall, these results indicate that our distributional alignment mechanism effectively stabilizes the most relevant semantic predictions under adversarial perturbations, supporting its design for robust open-vocabulary recognition in ambiguous multi-label settings.

### 6.2. Cross-Dataset Zero-Shot Adversarial Robustness

To verify the effectiveness of our approach under single-label settings, we analyze results across 16 datasets (Table 2). Our method achieves consistently strong performance, ranking best or second-best in nearly all cases. When trained with a single PGD regime, it generalizes effectively to multiple adversarial attacks while maintaining both the highest clean accuracy and adversarial robustness. The only ex-

ceptions are two domain-specific datasets (PCAM (Veeling et al., 2018) and EuroSAT (Helber et al., 2019)), which exhibit the highest predictive uncertainty (high PU in Fig. 1b, high AU in Fig. 6, and low EU in Fig. 7) and strong semantic overlap. These characteristics reflect a substantial departure from CLIP's natural-image pre-training domain, resulting in inherently weaker clean semantic geometry. Consequently, UCAT has less reliable structure to preserve through Dirichlet alignment, naturally limiting the magnitude of improvement. Nevertheless, UCAT remains stable on these domain-shifted datasets and achieves state-of-the-art robustness on the majority of natural-image benchmarks, where CLIP provides strong clean semantic structure and our Dirichlet alignment is most effective. We further extend our evaluation to larger-scale training, stronger attack settings, and additional ablations, with results reported in Appendix D.

### 6.3. Generalization across Vision–Language Backbones

We further evaluate whether UCAT generalizes beyond CLIP-B/32 to other *contrastively pretrained* VLMs, including CLIP-B/16 (Radford et al., 2021) and SLIP-B/16 (Mu et al., 2022) (Self-supervised Language–Image Pre-training). As shown in Table 3, UCAT consistently improves AutoAttack robustness and the clean–robust trade-off (harmonic mean, $H$) across backbones, indicating that our uncertainty-calibrated Dirichlet distribution matching is not tied to a specific CLIP variant but broadly applicable to contrastive

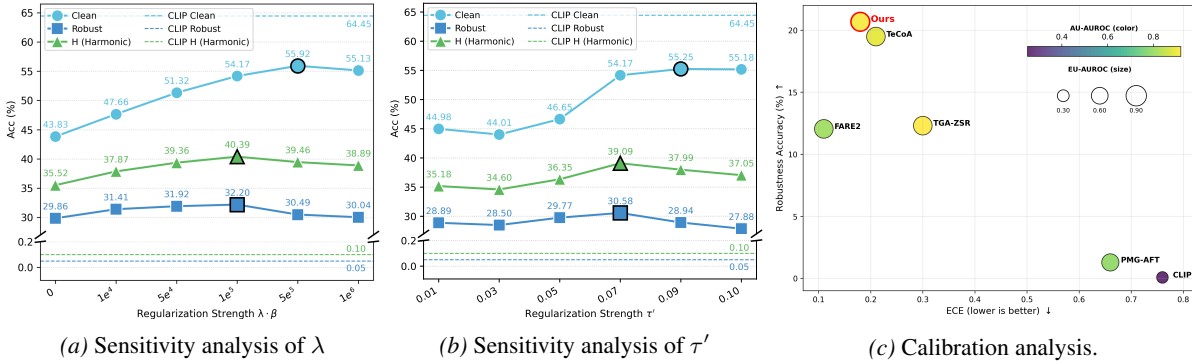

*(a) Sensitivity analysis of $\lambda$*      *(b) Sensitivity analysis of $\tau'$*      *(c) Calibration analysis.*

*Figure 3.* **Parameter sensitivity and robustness–calibration trade-off. (a) Sensitivity of regularization strength $\lambda$.** We vary $\lambda \cdot \beta \in \{10^4, 5 \times 10^4, 10^5, 5 \times 10^5, 10^6\}$ with $\beta = 2/e^{\tau'}$ and $\tau' = 0.07$. Models are adversarially fine-tuned with $\ell_\infty$ PGD-2 ($\epsilon = 1/255$) and evaluated by $\ell_\infty$ PGD-100 (Madry et al., 2017) (same $\epsilon$). Curves report averages over 16 datasets of Clean, Robust, and their harmonic mean $H$. **(b) Sensitivity of calibration coefficient $\tau'$.** Same as (a) while varying $\tau'$. **(c) Strong-attack robustness vs. calibration.** Models are fine-tuned with an $\ell_\infty$ 10-step PGD attack ($\epsilon = 2/255$) and evaluated with AutoAttack (Croce & Hein, 2020) at the same $\epsilon$, with all points averaged over 16 datasets. The x-axis: Expected Calibration Error (ECE; lower is better) and the y-axis is robust accuracy (higher is better). Bubble color/size encode AU-/EU-AUROC (aleatoric/epistemic uncertainty for error detection).

*Table 3.* **Generalization across contrastive VLM backbones.** We report the mean clean accuracy, AutoAttack robustness ($\ell_\infty$, $\epsilon = 1/255$), and harmonic mean $H$ over 16 datasets when fine-tuning different contrastively pretrained VLMs on TinyImageNet (2-step PGD, $\epsilon = 1/255$). Improvements of UCAT over the corresponding base model are shown in violet.

| Backbone | Method | Clean | AutoAttack | $H$ |
|---|---|---|---|---|
| CLIP-B/16 (Radford et al., 2021) | Base | 63.72 | 0.01 | 0.02 |
|  | +UCAT | 52.91 | 30.54 (+30.53) | 39.05 (+39.03) |
| CLIP-B/32 (Radford et al., 2021) | Base | 64.42 | 5.58 | 10.28 |
|  | +UCAT | 54.17 | 30.58 (+25.00) | 39.09 (+28.81) |
| SLIP-B/16 (Mu et al., 2022) | Base | 46.03 | 0.02 | 0.04 |
|  | +UCAT | 38.37 | 20.40 (+20.38) | 26.68 (+26.64) |

*Table 4.* **Ablation study.** Trained on TinyImageNet with 1-step PGD and evaluated under 100-step PGD, CW, and AutoAttack (AA) with $\epsilon = 1/255$. Results are averaged over 16 datasets. Best and second-best are in **bold** and underline.

| Methods | Clean | PGD | CW | AA |
|---|---|---|---|---|
| CLIP | 64.45 | 0.05 | 4.06 | 0.51 |
| $\mathcal{L}_{ce}$ | 43.83 | 29.86 | 29.25 | 28.74 |
| $\mathcal{L}_{ce} + \text{KL}(p(x)\|p(x^a))$ | 45.03 | 30.12 | 29.61 | 29.13 |
| $\mathcal{L}_{ce} + \text{KL}(p(x^a)\|p(x))$ | 45.05 | 29.98 | 29.28 | 28.80 |
| $\mathcal{L}_{ce} + \text{KL}(\text{Dir}(\alpha)\|\text{Dir}(\alpha_{adv}))$ | 36.72 | 25.01 | 24.66 | 24.36 |
| $\mathcal{L}_{ce} + \text{KL}(\text{Dir}(\alpha_{adv})\|\text{Dir}(\alpha))$ | **54.17** | **32.20** | **31.41** | **30.58** |

VLMs with shared image–text embedding spaces. Full per-dataset results are provided in Appendix D, Table 9.

## 6.4. Ablation and Parameter Sensitivity

Table 4 ablates our loss design. Starting from text-guided cross-entropy $\mathcal{L}_{ce}$, we systematically evaluate two key design dimensions: the distribution levels (probability vs. Dirichlet) and the KL divergence directions (forward vs. reverse). At the probability level (rows 3–4), introducing a KL term to align *softmax* distributions yields only a modest gain. This limited improvement stems from the fact that soft-

max normalization discards overall magnitude information, failing to preserve absolute evidence strength. When shifting to the Dirichlet level, comparing rows 5 and 6 clearly reveals that the reverse KL direction significantly outperforms its forward counterpart, validating the necessity of its mode-seeking behavior under adversarial perturbations. Consequently, our adopted reverse Dirichlet KL alignment (last row) simultaneously integrates both optimal choices to align both relative geometry and evidence magnitude, yielding the best observed trade-off between clean and robust performance.

Performance remains stable across a broad range of $\lambda$ values, with the best trade-off achieved at $10^5/\beta$ (Fig. 3a). Sensitivity to $\tau'$ is also mild around the default choice (Fig. 3b). Detailed results are provided in Appendix D.4.

## 6.5. Robustness and Calibration under Strong Attacks

Figure 3c provides a comprehensive evaluation under AutoAttack (Croce & Hein, 2020) with $\epsilon = 2/255$. We report four complementary metrics. *Expected Calibration Error (ECE)* (x-axis) measures how well predicted confidence matches actual correctness (lower is better), while *robustness accuracy* (y-axis) captures the ability to resist adversarial perturbations (higher is better). Bubble color denotes *AU-AUROC*, reflecting how aleatoric uncertainty helps identify errors caused by class ambiguity, and bubble size denotes *EU-AUROC*, reflecting how epistemic uncertainty captures errors due to insufficient evidence. An ideal model should lie toward the top-left of the plot (high robustness, low ECE) with large and bright bubbles (high AU-AUROC and EU-AUROC). Our method is closest to this desirable region: it achieves the highest robustness accuracy, maintains lower calibration error than existing baselines, and exhibits

stronger uncertainty discrimination as shown by larger and brighter bubbles. This demonstrates that our uncertainty calibration not only strengthens adversarial robustness but also improves predictive reliability under attack.

# 7. Conclusion

In this paper, we identified that adversarial perturbations in zero-shot CLIP not only reduce accuracy but also often suppress predictive uncertainty, leading to severe miscalibration. To address this, we reformulated CLIP logits as Dirichlet concentration parameters, yielding a representation that preserves both semantic structure and confidence strength. Building on this foundation, we introduced an uncertainty calibration adversarial finetuning method that aligns the Dirichlet distributions of clean and perturbed samples, ensuring robustness preservation and calibrated uncertainty. Extensive experiments demonstrate that our approach improves adversarial robustness, handles data ambiguity, and provides reliable uncertainty estimates. Beyond CLIP, our contrastive-theoretic perspective suggests a principled way to analyze and extend uncertainty modeling to other contrastive learning frameworks.

# Impact Statement

This work focuses on methodological advances in uncertainty modeling and robustness evaluation. The proposed approach is intended for research purposes and aims to improve understanding of model behavior under adversarial perturbations and multi-label ambiguity. While such insights may inform the design of more reliable learning systems, any deployment in real-world applications should be accompanied by thorough risk assessment and additional safeguards.

# Acknowledgment

This work was supported by the National Key R&D Program of China (No. 2023YFC2811500), the National Natural Science Foundation of China (No. 62272300), and the JSPS Bilateral Program (No. JPJSBP120257420).

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

# A. Implementation Details

**Datasets and evaluation suite.** We use the same zero-shot evaluation suite as prior zero-shot adversarial robustness (ZSAR) works (e.g., Mao et al. (2022)). Specifically, we evaluate on ImageNet/tinyImageNet(Deng et al., 2009), CIFAR10/100 (Krizhevsky et al., 2009), STL10 (Coates et al., 2011), Caltech101 (Fei-Fei et al., 2004), Caltech256 (Griffin et al., 2007), OxfordPets (Parkhi, Omkar M and Vedaldi, Andrea and Zisserman, Andrew and Jawahar, C. V., 2012), Stanford-Cars (Krause et al., 2013), Food101 (Bossard et al., 2014), Flowers102 (Nilsback & Zisserman, 2008), FGVC-Aircraft (Maji et al., 2013), SUN397 (Xiao et al., 2010), DTD (Cimpoi et al., 2014), and two domain-specialized sets PCAM (Veeling et al., 2018) and EuroSAT (Helber et al., 2019). To further assess robustness under semantic ambiguity, we additionally include the multi-label dataset MS-COCO (Lin et al., 2014).

**Backbone and training setup.** We adopt CLIP-B/32 (Radford et al., 2021) as the backbone and follow TeCoA's optimizer and training schedule (Mao et al., 2022), using a batch size of 256 and 10 training epochs unless otherwise stated. We benchmark five methods: CLIP (Radford et al., 2021), TeCoA (Mao et al., 2022), FARE (Schlarmann et al., 2024), PMG-AFT (Wang et al., 2024), TGA-ZSR (Yu et al., 2024), and Comp-TGA (Yu et al., 2026).

**Adversarial attacks.** For *training-time attacks,* we adopt two regimes: (i) a light regime following TeCoA (Mao et al., 2022), using $\ell_\infty$ PGD-2 with $\varepsilon = 1/255$; and (ii) a stronger regime following FARE (Schlarmann et al., 2024), using $\ell_\infty$ PGD-10 with $\varepsilon = 2/255$, where both regimes maintain a constant step size of $\alpha = 1/255$. For *evaluation attacks,* robustness is further assessed using $\ell_\infty$ PGD-100 (Madry et al., 2017) (with the same $\varepsilon$ as the training regime and a fixed step size of $\alpha = 1/255$), CW-100 (Carlini & Wagner, 2017), and AutoAttack (Croce & Hein, 2020) (the rand version ensembling APGD-CE and APGD-DLR).

**Loss weights.** We set $\lambda = 10^5/\beta$ with $\beta = 2/e^{\tau'}$, where $\tau' = 0.07$ follows standard contrastive learning practices (Wu et al., 2018; He et al., 2020; Radford et al., 2021; Yeh et al., 2022). Here $\beta$ corresponds to the upper bound of the mapping function $h(\ell)$ that converts logits $\ell$ into non-negative evidence. Using this bound guarantees that $\lambda$ remains numerically stable across different temperature values, preventing uncontrolled scaling when $\tau'$ varies.

# B. Proofs

### B.1. Proof of Lemma 4.3 (Validity of Dirichlet Evidence)

*Proof.* Since $\|v(x)\|_2 = \|t_k\|_2 = 1$, we have $\langle v(x), t_k \rangle \in [-1, 1]$. By the logit definition, $\tau \ell_k^{v \to t}(x) = \langle v(x), t_k \rangle \in [-1, 1]$. Therefore,

$$h(\ell_k^{v \to t}(x)) = \frac{\tau \, \ell_k^{v \to t}(x) + 1}{\tau'} \in \left[ 0, \, \frac{2}{\tau'} \right].$$

Exponentiating yields

$$\alpha_k(x) = \exp\big(h(\ell_k^{v \to t}(x))\big) \in \left[ e^0, \, e^{2/\tau'} \right] = \left[ 1, \, \exp(2/\tau') \right],$$

and both endpoints are attainable when $\langle v(x), t_k \rangle = -1$ and $+1$, respectively.

For monotonicity, differentiate $\alpha_k(x)$ with respect to $\ell_k^{v \to t}(x)$:

$$\frac{d \, \alpha_k(x)}{d \, \ell_k^{v \to t}(x)} = \frac{\tau}{\tau'} \exp\Big( \frac{\tau \, \ell_k^{v \to t}(x) + 1}{\tau'} \Big) = \frac{\tau}{\tau'} \alpha_k(x) > 0,$$

since $\tau > 0$, $\tau' > 0$, and $\alpha_k(x) > 0$. Hence $\alpha_k$ is strictly increasing in $\ell_k^{v \to t}$, which preserves both strict and non-strict order between any pair of logits. □

### B.2. Proof of Lemma 4.5 (Exact Equivalence at $\tau = \tau'$)

*Proof.* From the definition of the Dirichlet expectation in Equation 5,

$$p_k^{\text{Dir}}(x) = \mathbb{E}_{\pi \sim \text{Dir}(\alpha(x))}[\pi_k] = \frac{\alpha_k(x)}{\alpha_0(x)}, \quad \alpha_0(x) = \sum_{j=1}^C \alpha_j(x).$$

By construction,

$$\alpha_k(x) = \exp(h(\ell_k^{v \to t}(x))), \quad h(\ell_k^{v \to t}(x)) = \frac{\tau \ell_k^{v \to t}(x) + 1}{\tau'} = \frac{1}{\tau'} + \frac{\tau}{\tau'} \ell_k^{v \to t}(x).$$

Let $s = \tau/\tau' > 0$. Then

$$p_k^{\text{Dir}}(x) = \frac{\exp(1/\tau' + s\,\ell_k^{v \to t}(x))}{\sum_{j=1}^{C} \exp(1/\tau' + s\,\ell_j^{v \to t}(x))} = \frac{\exp(s\,\ell_k^{v \to t}(x))}{\sum_{j=1}^{C} \exp(s\,\ell_j^{v \to t}(x))} = \text{softmax}(s\,\ell^{v \to t}(x))_k,$$

since the additive constant $1/\tau'$ cancels out. When $s = 1$ (equivalently, $\tau' = \tau$), this reduces to

$$p_k^{\text{Dir}}(x) = \text{softmax}(\ell^{v \to t}(x))_k,$$

which matches exactly the original CLIP prediction $p_k^{\text{CLIP}}(x)$.

$\square$

### B.3. Proof of Corollary 4.7 (General form and invariances)

*Proof.* For any logits $\ell \in \mathbb{R}^C$ and scalar $s > 0$,

$$\arg\max_k \ell_k = \arg\max_k s\ell_k.$$

Since the softmax assigns the maximum probability to the index with maximum input, we have

$$\arg\max_k p_k^{\text{CLIP}}(x) = \arg\max_k p_k^{\text{Dir}}(x).$$

Thus both distributions yield the same classification decision, proving the accuray invariance.

For calibaration control, observe that $p_k^{\text{Dir}}(x) = e^{s\ell_k} / \sum_j e^{s\ell_j}$ becomes increasingly peaked as $s \to \infty$, converging to a one-hot vector, and tends to the uniform distribution as $s \to 0^+$. The entropy

$$H(p^{\text{Dir}}(x)) = -\sum_k p_k^{\text{Dir}}(x) \log p_k^{\text{Dir}}(x)$$

decreases monotonically with $s$. Thus $s$ leaves classification accuracy unchanged while directly modulating the calibration of predictive confidence. $\square$

## C. Extended Uncertainty Analysis

### C.1. Implementation Details for Uncertainty Quantification

Recall the decomposition of predictive uncertainty under the Dirichlet parameterization into aleatoric uncertainty (AU) and epistemic uncertainty (EU) in Section 3.3.

$$\text{AU}(x) = \mathbb{E}_{\pi \sim \text{Dir}(\alpha)}\big[H(\pi)\big] = -\sum_{k=1}^{C} \frac{\alpha_k}{\alpha_0}\Big(\psi(\alpha_k + 1) - \psi(\alpha_0 + 1)\Big), \quad \text{EU}(x) = \frac{C}{\alpha_0 + C}.$$

Our reformulation $\alpha_k(x) = \exp\big(h(\ell_k^{v \to t}(x))\big)$, $h(\ell) = \frac{\tau\,\ell + 1}{\tau'}$, adopts a linear definition of the evidence mapping $h(\ell)$, for which Section 4 and Appendix B have established the theoretical equivalence between CLIP logits and Dirichlet distributions.

In practice, however, the learnable temperature coefficient $\tau$ may become very small during training (e.g., $\tau = 0.01$), which leads to excessively large logits after exponentiation and renders the raw uncertainty values numerically unstable. To address this, we introduce an additional activation $h'(\ell) = \text{softplus}(h(\ell))$, which is commonly adopted in EDL to smooth the outputs and map them into a numerically stable range suitable for analysis (Sensoy et al., 2018; Malinin & Gales, 2018).

Moreover, when $\tau$ is too small (e.g., $\tau = 0.01$), EU degenerates towards 0 and AU coincides with PU. To avoid this issue, we adopt $\tau = 0.07$ for computing EU, while keeping $\tau = 0.01$ for AU. This choice is theoretically acceptable: both the softplus mapping and the rescaling by $\tau$ affect only the magnitude of uncertainty values, not their ordering. As a result, the reliability of AUROC evaluation, which depends only on ranking, is unaffected. For ECE, we use PU directly computed from probabilities, which is independent of $\tau$ and activation adjustments.

These practical adjustments ensure stable and meaningful AU/EU quantification without altering the comparative reliability of our uncertainty metrics.

## C.2. Additional Visualizations of Uncertainty

To complement the main results, we provide extended visualizations that follow the analysis chain from *overall predictive uncertainty* to its *aleatoric/epistemic* decomposition.

**Attack-wise robustness vs. predictive uncertainty shift.** Figure 4 reports the joint change in accuracy and predictive uncertainty (PU) under three strong white-box attacks (PGD, CW, AutoAttack). The consistently negative $\Delta$PU for CLIP indicates that miscalibration is not tied to a specific attack, motivating uncertainty-aware regularization.

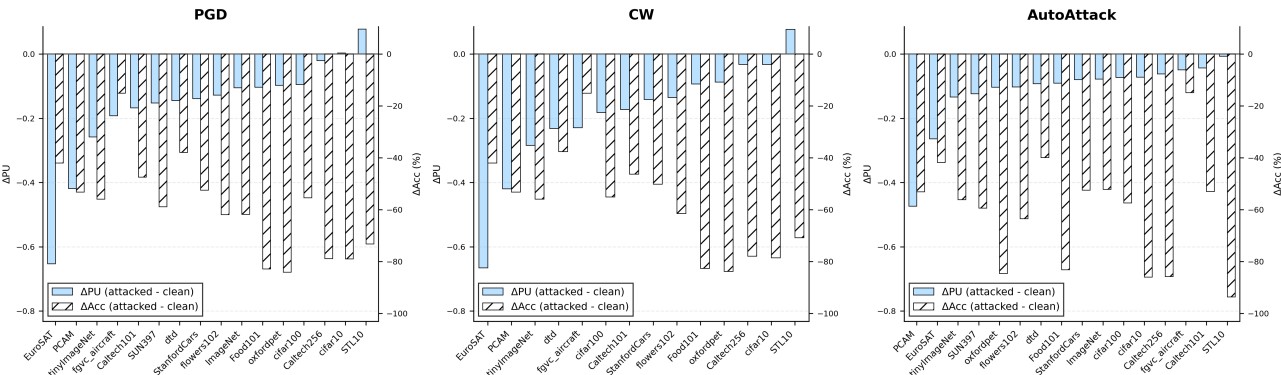

*Figure 4.* Effect of strong white-box attacks ($\epsilon = 1/255$, 100 steps) on accuracy and predictive uncertainty across 16 datasets. Each panel shows the change under a single attack type (left: PGD, center: CW, right: AutoAttack); for each dataset the filled light bars plot $\Delta$PU = $\text{PU}_{\text{attacked}} - \text{PU}_{\text{clean}}$ (left axis) and the hatched bars plot $\Delta$Acc = $\text{Acc}_{\text{attacked}} - \text{Acc}_{\text{clean}}$ in percentage points (right axis). Negative values therefore indicate decreases caused by the attack. Results demonstrate that all three attacks induce simultaneous drops in accuracy and predictive uncertainty on most datasets, with the magnitude of degradation varying by dataset and attack.

**Predictive uncertainty (PU).** Figure 5 summarizes PU (entropy) on 16 datasets for clean and adversarial inputs. Across most datasets, CLIP exhibits *uncertainty suppression* under attack (lower entropy on adversarial inputs), which manifests as over-confident yet incorrect predictions. Our method mitigates this effect and yields a more consistent ordering between clean and adversarial uncertainty by explicitly regularizing evidence scale in addition to relative class structure.

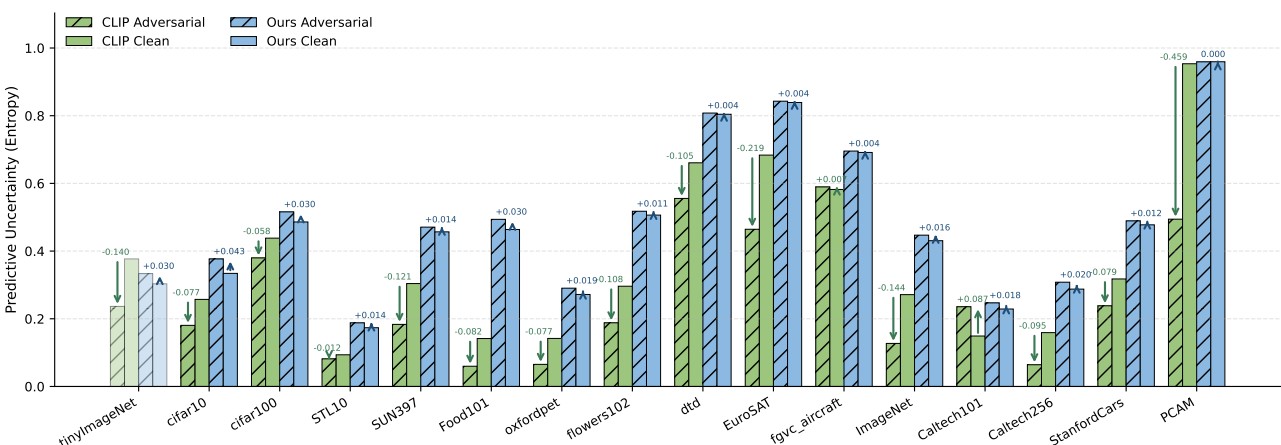

*Figure 5.* **Predictive uncertainty on 16 datasets**. CLIP shows reduced entropy on adversarial inputs, whereas our method UCAT restores calibrated uncertainty. Arrows and numbers show uncertainty change (direction, magnitude).

**Decomposition into AU/EU.** Finally, Figures 6 and 7 decompose uncertainty into aleatoric (AU) and epistemic (EU) components. Together, these plots show that adversarial perturbations can distort both class ambiguity (AU) and evidence strength (EU), whereas our method yields more reliable AU/EU behaviors across datasets.

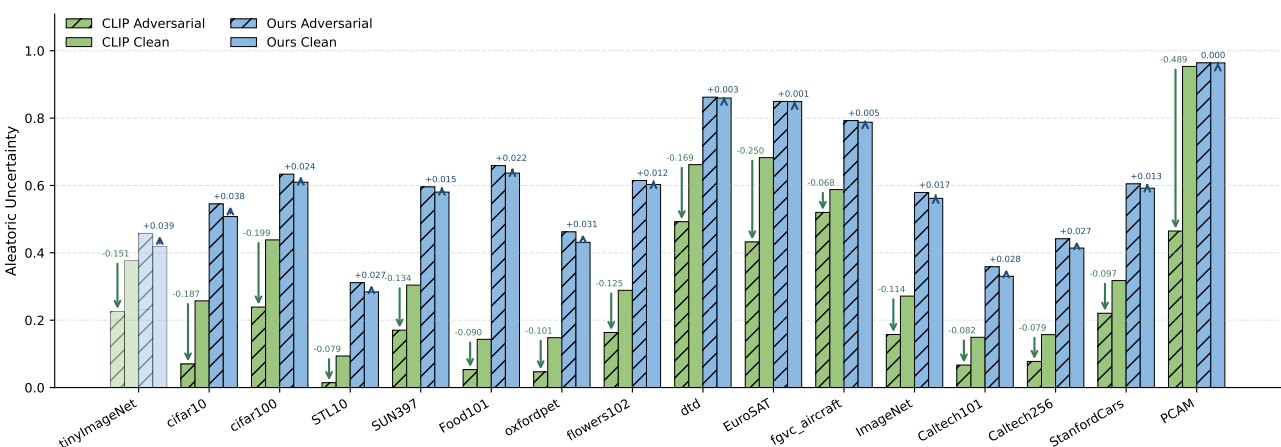

*Figure 6.* Comparison of **aleatoric uncertainty** on clean and adversarial samples across 16 datasets between CLIP and our method, adversarially trained on tinyImageNet under 10-step PGD with $\epsilon = 2/255$.

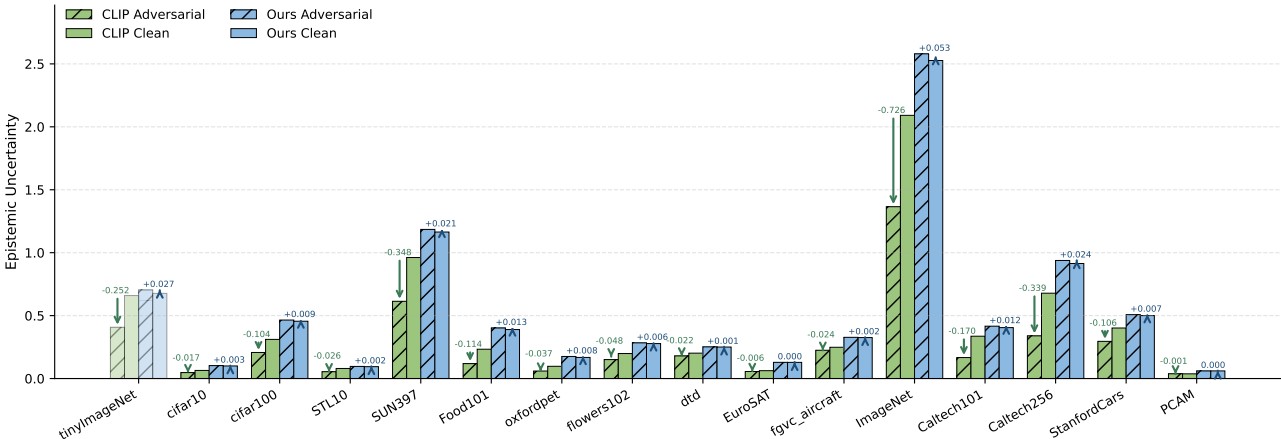

*Figure 7.* Comparison of **epistemic uncertainty** on clean and adversarial samples across 16 datasets between original CLIP and our method, adversarially trained on tinyImageNet under 10-step PGD with $\epsilon = 2/255$.

## D. Evaluation under Stronger Attacks, Larger Datasets, and Additional Ablations

### D.1. Multi-target PGD evaluation on multi-label MS-COCO

Here we additionally evaluate a stronger *multi-label* PGD attack on MS-COCO. Specifically, instead of attacking a single label at a time, we jointly maximize `BCEWithLogitsLoss` over all target labels, resulting in a *joint multi-objective* adversarial perturbation that simultaneously degrades multiple semantic predictions. As reported in Table 5, UCAT remains competitive and achieves the best (or tied-best) clean–robust trade-off in $H(\mathrm{F1@3})$ under moderate-to-large perturbation budgets (e.g., $\epsilon \in 2/255, 4/255$), suggesting that distributional (Dirichlet) alignment better preserves a stable multi-label neighborhood under stronger joint adversarial perturbations.

### D.2. Training under stronger adversarial attacks

TeCoA (Mao et al., 2022) and FARE (Schlarmann et al., 2024) represent two widely used adversarial fine-tuning configurations on TinyImageNet, with FARE adopting a substantially stronger perturbation budget. To ensure a fair and comprehensive comparison, we evaluate UCAT under this stronger training regime as well. Specifically, we follow the FARE configuration and train models using 10-step PGD with $\epsilon = 2/255$.

*Table 5.* **Zero-shot adversarial robustness on multi-label dataset MS-COCO (Lin et al., 2014).** All ZSAR models are adversarially trained on TinyImageNet with the FARE (Schlarmann et al., 2024) 10-step PGD setting ($\epsilon = 1/255$), and evaluated under **PGD**-100 at radii $\epsilon \in \{1/255, 2/255, 4/255\}$ plus clean. We report mean Average Precision (mAP), Precision (P), Recall (R), and F1-score (F1) at top-3 predictions. $H(\text{F1@3})$ denotes the harmonic mean of clean and adversarial F1@3. Best and second-best are in **bold** and underline.

| | Clean | | | | $\epsilon = 1/255$ | | | | | $\epsilon = 2/255$ | | | | | $\epsilon = 4/255$ | | | | |
| Methods | mAP | P@3 | R@3 | F1@3 | mAP | P@3 | R@3 | F1@3 | $H_{(F1@3)}$ | mAP | P@3 | R@3 | F1@3 | $H_{(F1@3)}$ | mAP | P@3 | R@3 | F1@3 | $H_{(F1@3)}$ |
|---|---|---|---|---|---|---|---|---|---|---|---|---|---|---|---|---|---|---|---|
| CLIP (Radford et al., 2021) | 51.96 | 45.33 | 46.49 | 45.90 | 40.21 | 40.43 | 41.45 | 40.93 | 43.27 | 24.94 | 27.85 | 28.54 | 28.19 | 34.93 | 11.30 | 15.55 | 15.94 | 15.74 | 23.44 |
| TeCoA (Mao et al., 2022) | 46.34 | 33.73 | 34.57 | 34.14 | 38.90 | 39.14 | 40.14 | 39.63 | 36.68 | 42.10 | 30.54 | 31.32 | 30.92 | 32.45 | 31.29 | 25.55 | 26.21 | 25.87 | 29.44 |
| FARE (Schlarmann et al., 2024) | 49.21 | 43.83 | 44.95 | 44.37 | 43.33 | 38.63 | 39.61 | 39.11 | 41.57 | 23.82 | 32.89 | 33.74 | 33.30 | 38.05 | 11.21 | 24.31 | 24.93 | 24.61 | 31.66 |
| PMG-AFT (Wang et al., 2024) | 48.94 | 41.77 | 42.83 | 42.29 | 46.42 | 36.00 | 36.91 | 36.44 | 39.15 | 34.56 | 32.91 | 33.75 | 33.32 | 37.27 | 15.51 | 25.16 | 25.81 | 25.47 | 31.79 |
| TGA-ZSR (Yu et al., 2024) | 48.61 | 37.19 | 38.13 | 37.65 | 44.62 | 32.33 | 33.15 | 32.73 | 35.02 | 42.79 | 33.52 | 34.37 | 33.93 | 35.69 | 29.20 | 27.31 | 28.01 | 27.65 | 31.88 |
| Comp-TGA (Yu et al., 2026) | 48.14 | 38.11 | 39.07 | 38.58 | 44.52 | 36.10 | 37.01 | 36.54 | 37.53 | 40.07 | 33.32 | 34.17 | 33.73 | 35.99 | 25.16 | 26.97 | 27.66 | 27.30 | 31.97 |
| UCAT (Ours) | 47.14 | 41.55 | 42.62 | 42.07 | 44.31 | 40.10 | 41.12 | 40.60 | 41.32 | 39.88 | 37.47 | 38.43 | 37.94 | 39.90 | 28.01 | 29.95 | 30.72 | 30.32 | 35.24 |

*Table 6.* **Zero-shot adversarial robustness under 10-step PGD training.** All methods are fine-tuned on TinyImageNet following FARE (Schlarmann et al., 2024), and adversarial finetuning uses 10-step PGD (Madry et al., 2017) with $\epsilon = 2/255$. We report zero-shot robustness across 16 single-label datasets under five different white-box and adaptive attacks (PGD (Madry et al., 2017), CW (Carlini & Wagner, 2017), AutoAttack (Croce & Hein, 2020), CAA (Mao et al., 2021) and $A^3$ (Liu et al., 2022)). $H$ is the harmonic mean between Clean and the corresponding robust score. Best and second-best are in **bold** and underline.

| | Methods | TinyImageNet | CIFAR-10 | CIFAR-100 | STL10 | SUN397 | Food101 | Oxfordpets | Flowers102 | DTD | EuroSAT | FGVC Aircraft | ImageNet | Caltech101 | Caltech256 | StanfordCars | PCAM | Average | H |
|---|---|---|---|---|---|---|---|---|---|---|---|---|---|---|---|---|---|---|---|
| Clean | CLIP (Radford et al., 2021) | 57.96 | 88.03 | 60.45 | 97.03 | 57.26 | 83.89 | 87.41 | 65.49 | 40.64 | 42.66 | 20.16 | 59.15 | 85.32 | 81.73 | 52.02 | 52.08 | 64.45 | |
| | TeCoA (Mao et al., 2022) | 63.20 | 58.62 | 31.75 | 80.59 | 25.71 | 19.15 | 49.25 | 24.61 | 17.34 | 15.89 | 2.88 | 24.70 | 63.04 | 47.67 | 13.11 | 49.97 | 36.72 | |
| | FARE (Schlarmann et al., 2024) | 16.92 | 40.23 | 11.96 | 64.56 | 7.89 | 8.07 | 19.24 | 11.82 | 7.93 | 12.52 | 2.55 | 7.98 | 47.14 | 27.61 | 6.06 | 50.02 | 21.41 | |
| | PMG-AFT (Wang et al., 2024) | 46.62 | 70.37 | 38.49 | 90.34 | 52.02 | 57.11 | 79.75 | 49.36 | 32.23 | 23.43 | 12.03 | 47.70 | 82.49 | 73.65 | 41.60 | 55.87 | 53.32 | |
| | TGA-ZSR (Yu et al., 2024) | 69.78 | 83.98 | 52.32 | 91.36 | 44.70 | 50.17 | 72.55 | 45.05 | 26.92 | 27.58 | 10.68 | 41.84 | 80.04 | 71.94 | 33.14 | 50.02 | 53.25 | |
| | Comp-TGA (Yu et al., 2026) | 66.58 | 82.25 | 49.40 | 91.35 | 41.73 | 46.89 | 70.13 | 44.14 | 26.01 | 26.96 | 8.16 | 38.31 | 78.85 | 70.33 | 32.63 | 50.02 | 51.48 | |
| | UCAT (Ours) | 67.18 | 66.52 | 41.07 | 86.73 | 30.10 | 36.97 | 62.66 | 36.69 | 24.95 | 19.39 | 7.26 | 32.61 | 75.00 | 60.15 | 26.39 | 49.66 | 45.21 | |
| PGD | CLIP (Radford et al., 2021) | 0.00 | 0.94 | 0.28 | 0.45 | 0.00 | 0.00 | 0.00 | 0.00 | 0.11 | 0.00 | 0.00 | 0.00 | 0.77 | 0.19 | 0.00 | 0.00 | 0.00 | 0.00 |
| | TeCoA (Mao et al., 2022) | 35.74 | 23.57 | 14.47 | 53.50 | 10.20 | 6.58 | 22.90 | 10.96 | 10.75 | 11.72 | 0.57 | 10.53 | 40.27 | 27.02 | 3.61 | 49.31 | 20.73 | 26.50 |
| | FARE (Schlarmann et al., 2024) | 8.62 | 18.19 | 5.67 | 41.10 | 3.46 | 2.88 | 7.01 | 5.22 | 5.21 | 9.29 | 0.96 | 3.47 | 31.98 | 15.47 | 1.73 | 50.02 | 13.14 | 16.29 |
| | PMG-AFT (Wang et al., 2024) | 7.08 | 13.18 | 3.63 | 55.79 | 3.39 | 7.90 | 12.81 | 4.73 | 5.11 | 4.47 | 0.00 | 4.38 | 39.65 | 26.22 | 0.44 | 0.94 | 11.86 | 19.40 |
| | TGA-ZSR (Yu et al., 2024) | 30.74 | 20.17 | 12.02 | 51.99 | 9.46 | 6.69 | 20.58 | 12.47 | 9.11 | 11.22 | 0.63 | 10.28 | 40.63 | 29.06 | 3.56 | 49.97 | 20.02 | 29.10 |
| | Comp-TGA (Yu et al., 2026) | 30.28 | 18.79 | 11.53 | 48.83 | 8.71 | 5.45 | 19.60 | 11.87 | 11.12 | 11.42 | 0.45 | 9.02 | 36.67 | 24.82 | 2.90 | 49.93 | 18.84 | 27.58 |
| | UCAT (Ours) | 35.38 | 25.81 | 15.67 | 58.44 | 11.48 | 11.17 | 26.82 | 15.04 | 13.94 | 4.53 | 1.20 | 13.13 | 51.34 | 34.60 | 6.72 | 34.02 | 22.45 | 30.01 |
| CW | CLIP (Radford et al., 2021) | 0.00 | 0.58 | 0.20 | 0.57 | 0.08 | 0.00 | 0.00 | 0.00 | 0.12 | 0.00 | 0.00 | 0.08 | 0.24 | 0.33 | 2.19 | 0.00 | 0.27 | 0.55 |
| | TeCoA (Mao et al., 2022) | 33.90 | 23.05 | 13.66 | 52.50 | 8.95 | 5.74 | 22.13 | 10.05 | 9.42 | 11.40 | 0.63 | 9.58 | 39.92 | 26.04 | 3.10 | 49.26 | 19.96 | 25.86 |
| | FARE (Schlarmann et al., 2024) | 7.04 | 14.64 | 4.63 | 38.94 | 2.91 | 2.20 | 6.68 | 4.38 | 4.36 | 8.43 | 0.75 | 2.96 | 30.13 | 14.24 | 1.73 | 50.02 | 12.13 | 15.48 |
| | PMG-AFT (Wang et al., 2024) | 1.64 | 0.27 | 0.79 | 14.04 | 1.89 | 1.15 | 2.51 | 3.09 | 4.42 | 5.83 | 0.00 | 1.88 | 19.58 | 11.41 | 2.02 | 1.03 | 4.47 | 8.25 |
| | TGA-ZSR (Yu et al., 2024) | 29.68 | 20.48 | 11.48 | 51.53 | 9.15 | 6.43 | 21.72 | 12.05 | 9.63 | 11.01 | 0.60 | 10.06 | 40.79 | 28.67 | 4.48 | 49.97 | 19.86 | 28.93 |
| | Comp-TGA (Yu et al., 2026) | 27.92 | 18.56 | 10.77 | 47.96 | 7.94 | 4.83 | 19.24 | 11.14 | 9.11 | 11.23 | 0.42 | 8.47 | 36.14 | 23.90 | 3.62 | 49.93 | 18.21 | 26.90 |
| | UCAT (Ours) | 34.64 | 25.46 | 14.69 | 57.88 | 10.25 | 9.83 | 26.49 | 12.91 | 11.70 | 3.80 | 1.11 | 11.95 | 50.47 | 33.40 | 6.36 | 33.09 | 21.50 | 29.14 |
| AutoAttack | CLIP (Radford et al., 2021) | 0.00 | 0.05 | 0.14 | 0.00 | 0.02 | 0.03 | 0.03 | 0.02 | 0.13 | 0.17 | 0.23 | 0.03 | 0.02 | 0.06 | 0.07 | 0.12 | 0.07 | 0.14 |
| | TeCoA (Mao et al., 2022) | 32.68 | 21.94 | 13.17 | 51.93 | 8.43 | 5.54 | 21.56 | 9.92 | 9.52 | 11.36 | 0.51 | 9.10 | 38.88 | 25.35 | 2.56 | 49.23 | 19.48 | 25.46 |
| | FARE (Schlarmann et al., 2024) | 7.00 | 14.81 | 4.58 | 38.71 | 2.81 | 2.16 | 6.49 | 4.29 | 4.47 | 8.52 | 0.72 | 2.86 | 29.84 | 14.03 | 1.34 | 50.02 | 12.04 | 15.41 |
| | PMG-AFT (Wang et al., 2024) | 0.62 | 0.14 | 0.25 | 7.58 | 0.49 | 0.40 | 0.55 | 0.94 | 2.29 | 0.04 | 0.15 | 0.56 | 12.33 | 6.19 | 0.04 | 0.37 | 2.06 | 3.96 |
| | TGA-ZSR (Yu et al., 2024) | 11.28 | 6.29 | 5.53 | 36.76 | 4.00 | 3.27 | 9.27 | 6.18 | 6.33 | 8.94 | 0.18 | 5.13 | 30.23 | 19.57 | 0.96 | 42.88 | 12.30 | 19.98 |
| | Comp-TGA (Yu et al., 2026) | 11.84 | 6.64 | 4.94 | 34.56 | 3.95 | 2.80 | 9.02 | 5.82 | 6.60 | 9.68 | 0.12 | 4.69 | 27.71 | 16.99 | 0.87 | 43.61 | 11.87 | 19.29 |
| | UCAT (Ours) | 32.84 | 24.08 | 13.97 | 57.15 | 9.50 | 9.09 | 24.72 | 12.60 | 11.97 | 3.76 | 0.78 | 11.11 | 49.44 | 32.35 | 4.71 | 32.62 | 20.67 | 28.37 |
| CAA | CLIP (Radford et al., 2021) | 1.90 | 5.84 | 0.32 | 28.99 | 0.63 | 9.80 | 5.04 | 1.29 | 0.05 | 0.04 | 0.00 | 1.02 | 16.63 | 13.17 | 0.42 | 0.00 | 5.32 | 9.83 |
| | TeCoA (Mao et al., 2022) | 35.52 | 23.40 | 14.38 | 53.25 | 9.98 | 6.49 | 22.65 | 10.95 | 10.69 | 11.62 | 0.57 | 10.33 | 40.03 | 26.83 | 3.43 | 49.28 | 20.59 | 26.38 |
| | FARE (Schlarmann et al., 2024) | 8.50 | 17.74 | 5.51 | 40.86 | 3.39 | 2.80 | 6.95 | 5.14 | 5.05 | 8.86 | 0.93 | 3.41 | 31.72 | 15.32 | 1.65 | 50.02 | 12.99 | 16.17 |
| | PMG-AFT (Wang et al., 2024) | 7.18 | 13.18 | 3.79 | 56.18 | 3.37 | 8.21 | 13.63 | 4.36 | 3.19 | 0.32 | 0.00 | 4.43 | 39.61 | 26.36 | 0.44 | 0.39 | 11.54 | 18.97 |
| | TGA-ZSR (Yu et al., 2024) | 18.64 | 10.77 | 8.41 | 42.23 | 5.78 | 4.51 | 11.72 | 7.81 | 7.87 | 10.54 | 0.27 | 6.73 | 33.71 | 22.49 | 1.62 | 42.37 | 14.72 | 23.06 |
| | Comp-TGA (Yu et al., 2026) | 10.60 | 7.58 | 5.59 | 38.86 | 4.86 | 3.37 | 9.02 | 6.34 | 7.39 | 8.88 | 0.09 | 5.65 | 29.60 | 18.75 | 1.43 | 41.75 | 12.48 | 20.10 |
| | **UCAT** | 35.12 | 25.69 | 15.42 | 58.33 | 11.16 | 10.91 | 26.90 | 14.73 | 13.83 | 4.33 | 1.08 | 12.75 | 51.11 | 34.33 | 6.29 | 33.45 | 22.21 | 29.79 |
| $A^3$ | CLIP (Radford et al., 2021) | 1.26 | 6.47 | 0.33 | 30.70 | 0.70 | 9.73 | 4.80 | 1.11 | 0.11 | 0.10 | 0.00 | 1.00 | 19.11 | 13.56 | 0.42 | 0.00 | 5.58 | 10.28 |
| | TeCoA (Mao et al., 2022) | 35.38 | 23.21 | 14.22 | 53.11 | 9.95 | 6.46 | 22.57 | 10.83 | 10.59 | 11.62 | 0.57 | 10.29 | 39.98 | 26.75 | 3.47 | 49.27 | 20.52 | 26.32 |
| | FARE (Schlarmann et al., 2024) | 8.46 | 17.55 | 5.49 | 40.65 | 3.39 | 2.78 | 6.87 | 5.12 | 5.05 | 8.89 | 0.93 | 3.40 | 31.69 | 15.31 | 1.65 | 50.02 | 12.95 | 16.14 |
| | PMG-AFT (Wang et al., 2024) | 7.02 | 13.16 | 3.65 | 55.74 | 3.37 | 7.90 | 12.87 | 4.68 | 4.95 | 4.45 | 0.00 | 4.37 | 39.60 | 26.18 | 0.44 | 0.89 | 11.83 | 19.36 |
| | TGA-ZSR (Yu et al., 2024) | 30.02 | 19.54 | 11.59 | 51.60 | 9.02 | 6.44 | 19.98 | 12.21 | 10.64 | 11.13 | 0.54 | 9.85 | 40.07 | 28.50 | 3.20 | 49.95 | 19.64 | 28.70 |
| | Comp-TGA (Yu et al., 2026) | 29.20 | 18.24 | 11.13 | 48.39 | 8.28 | 5.29 | 18.89 | 11.58 | 10.37 | 11.26 | 0.42 | 8.70 | 36.28 | 24.42 | 2.72 | 49.93 | 18.44 | 27.16 |
| | UCAT | 34.90 | 25.44 | 15.33 | 58.14 | 11.08 | 10.82 | 26.41 | 14.67 | 13.83 | 4.33 | 1.08 | 12.69 | 51.02 | 34.22 | 6.19 | 33.34 | 22.09 | 29.68 |

We first report zero-shot robustness under three standard white-box attacks (*i.e.*, PGD, CW, and AutoAttack) in Table 6, which measures performance against substantially stronger perturbations than those used in Table 2. To further assess generalization across attack types, we additionally evaluate CAA (Mao et al., 2021) and $A^3$ (Liu et al., 2022), two strong adaptive attacks. Together, these evaluations examine UCAT's robustness under both stronger training perturbations and a wider range of test-time threat models.

## D.3. Training on a larger dataset

Following TeCoA (Mao et al., 2022), we train on ImageNet-1k with 2-step PGD at $\epsilon = 1/255$ to assess performance on a larger training dataset across 15 benchmarks (tinyImageNet is excluded, as it was not reported in TeCoA's original paper). As shown in Table 7, this setting examines whether UCAT continues to benefit from its uncertainty-calibration mechanism when trained on large-scale data.

*Table 7.* **Zero-shot adversarial robustness across 15 datasets.** All methods are fine-tuned on ImageNet-1k following TeCoA (Mao et al., 2022), adversarial training uses 2-step PGD (Madry et al., 2017) with $\epsilon=1/255$. *Average* is the mean across datasets; *H* is the harmonic mean between Clean and the corresponding robust score. Best and second-best are in **bold** and underline.

| | Methods | CIFAR-10 | CIFAR-100 | STL10 | SUN397 | Food101 | Oxfordpets | Flowers102 | DTD | EuroSAT | FGVC Aircraft | ImageNet | Caltech101 | Caltech256 | StanfordCars | PCAM | Average | H |
|---|---|---|---|---|---|---|---|---|---|---|---|---|---|---|---|---|---|---|
| Clean | CLIP (Radford et al., 2021) | 88.03 | 60.45 | 97.03 | 57.26 | 83.89 | 87.41 | 65.49 | 40.64 | 42.66 | 20.16 | 59.15 | 85.32 | 81.73 | 52.02 | 52.08 | 64.89 | |
| | TeCoA (Mao et al., 2022) | 78.12 | 49.68 | 93.30 | 51.28 | 55.37 | 81.58 | 50.92 | 34.15 | **27.57** | 13.89 | 63.87 | 83.51 | 76.51 | 33.30 | **49.01** | 56.14 | |
| | FARE (Schlarmann et al., 2024) | **84.75** | **59.85** | **95.69** | 53.97 | 75.58 | **86.92** | **60.48** | 36.86 | 24.74 | **17.10** | **85.01** | **85.01** | **80.57** | **49.71** | 45.06 | **62.75** | |
| | UCAT (Ours) | 83.78 | 58.11 | 95.65 | 53.98 | 68.84 | 86.05 | 58.30 | **37.18** | 23.02 | 15.24 | 70.48 | 84.64 | 80.27 | 44.96 | 46.56 | 60.47 | |
| Auto Attack | CLIP (Radford et al., 2021) | 9.57 | 4.55 | 35.40 | 1.02 | 3.95 | 2.72 | 1.19 | 2.50 | 0.04 | 0.00 | 1.72 | 24.63 | 7.19 | 0.27 | 0.10 | 0.05 | 0.10 |
| | TeCoA (Mao et al., 2022) | **59.28** | 34.13 | **83.45** | **29.81** | 27.99 | 62.61 | 30.69 | 22.88 | **15.18** | 5.10 | **41.88** | **69.07** | 59.54 | 13.37 | 23.87 | **38.59** | 45.74 |
| | FARE (Schlarmann et al., 2024) | 50.96 | **28.48** | 80.88 | 26.66 | **34.36** | 61.43 | **31.91** | 24.31 | 14.12 | **5.28** | 32.11 | 68.19 | 59.95 | 18.52 | 25.74 | 37.53 | 46.97 |
| | UCAT (Ours) | 50.59 | **28.48** | 82.09 | 29.93 | 33.72 | 67.59 | 33.26 | 24.42 | 12.65 | 5.73 | 47.51 | 71.11 | 62.71 | 19.62 | 25.84 | 39.68 | 47.92 |
| PGD | CLIP (Radford et al., 2021) | 2.54 | 1.11 | 3.18 | 0.05 | 0.03 | 0.02 | 0.03 | 0.17 | 0.23 | 0.04 | 0.10 | 0.26 | 0.07 | 0.07 | 0.12 | 0.54 | 1.08 |
| | TeCoA (Mao et al., 2022) | 58.27 | 32.57 | 83.16 | 29.03 | 25.79 | 61.76 | 28.93 | 20.70 | 13.26 | 4.05 | 48.51 | 68.40 | 58.59 | 12.03 | 24.09 | 37.94 | 45.28 |
| | FARE (Schlarmann et al., 2024) | 49.62 | 25.98 | 80.60 | 24.77 | 33.06 | 60.51 | 29.55 | 22.02 | 12.95 | 4.08 | 39.81 | 67.21 | 58.87 | 16.43 | 25.56 | 36.73 | 46.34 |
| | UCAT (Ours) | 49.00 | 26.42 | 81.73 | 27.85 | 31.88 | 66.86 | 30.64 | 22.45 | 10.76 | 4.50 | 45.59 | 70.12 | 61.64 | 17.40 | 25.37 | 38.15 | 46.78 |

## D.4. Effect of the calibration coefficient $\tau'$

As shown in Table 8, We perform a controlled ablation by varying the Dirichlet calibration coefficient $\tau'$ and reporting clean accuracy, AutoAttack robustness, and their harmonic mean $H$ across 16 datasets. This study isolates the influence of the evidence-scaling term in our formulation and identifies $\tau' = 0.07$ as the best operating point.

*Table 8.* **Effect of the calibration coefficient $\tau'$ on zero-shot adversarial robustness across 16 single-label datasets.** All methods are fine-tuned on TinyImageNet following TeCoA (Yu et al., 2024), adversarial training uses 2-step PGD (Madry et al., 2017) with $\epsilon=1/255$. *Average* is the mean across datasets. *H* is the harmonic mean between Clean and the corresponding robust score. Best and second-best are in **bold** and underline.

| | Methods | TinyImageNet | CIFAR-10 | CIFAR-100 | STL10 | SUN397 | Food101 | Oxfordpets | Flowers102 | DTD | EuroSAT | FGVC Aircraft | ImageNet | Caltech101 | Caltech256 | StanfordCars | PCAM | Average | H |
|---|---|---|---|---|---|---|---|---|---|---|---|---|---|---|---|---|---|---|---|
| Clean | CLIP (Radford et al., 2021) | 57.96 | 88.02 | 60.47 | 97.03 | 57.26 | 83.89 | 87.38 | 65.52 | 40.69 | 42.65 | 20.16 | 59.15 | 85.33 | 81.73 | 51.98 | 52.08 | 64.46 | |
| | $\tau' = 0.01$ | 70.06 | 70.33 | 39.38 | 86.74 | 37.22 | 31.12 | 63.78 | 35.16 | 25.43 | 16.87 | 4.89 | 34.23 | 73.42 | 60.46 | 20.56 | 49.99 | 44.98 | |
| | $\tau' = 0.03$ | 69.38 | 64.18 | 36.23 | 86.56 | 36.74 | 31.13 | 62.50 | 35.52 | 25.75 | 14.24 | 5.13 | 33.86 | 73.30 | 59.63 | 20.15 | 49.92 | 44.01 | |
| | $\tau' = 0.05$ | 72.68 | 71.18 | 42.52 | 88.33 | 36.36 | 31.31 | 66.78 | 37.10 | 26.44 | 16.86 | 5.85 | 35.98 | 76.29 | 61.66 | 24.69 | 49.83 | 46.65 | |
| | $\tau' = 0.07$ **(Ours)** | 74.46 | 81.81 | 54.45 | 91.88 | 41.06 | 53.58 | 74.16 | 47.57 | 31.92 | **19.29** | 10.95 | 43.20 | **82.39** | 71.53 | 37.32 | 51.20 | 54.17 | |
| | $\tau' = 0.09$ | 70.66 | 83.79 | 56.44 | 92.23 | 42.10 | 58.26 | 75.88 | 49.16 | 33.56 | 16.87 | 11.67 | 43.62 | 82.07 | 72.96 | 38.48 | 56.23 | 55.25 | |
| | $\tau' = 0.10$ | 70.40 | **85.27** | **57.27** | 92.16 | **42.51** | 57.85 | 76.34 | 48.97 | 32.93 | 15.28 | 11.19 | 43.56 | 81.78 | 72.90 | 38.43 | 56.08 | 55.18 | |
| Auto Attack | CLIP (Radford et al., 2021) | 1.26 | 6.47 | 0.33 | 30.70 | 0.70 | 9.73 | 4.80 | 1.11 | 0.11 | 0.10 | 0.00 | 1.00 | 19.11 | 13.56 | 0.37 | 0.00 | 5.58 | 10.28 |
| | $\tau' = 0.01$ | 45.98 | 37.00 | 20.52 | 69.35 | 18.07 | 12.52 | 37.80 | 20.31 | 16.86 | **11.53** | 1.74 | 17.21 | 54.62 | 41.67 | 7.82 | **49.16** | 28.89 | 35.18 |
| | $\tau' = 0.03$ | 46.30 | 33.71 | 18.71 | 68.74 | 18.08 | 13.20 | 37.78 | 20.61 | 16.70 | 11.29 | 1.77 | 17.41 | 55.72 | 41.57 | 7.46 | 46.95 | 28.50 | 34.60 |
| | $\tau' = 0.05$ | **48.48** | 37.96 | 21.59 | 70.98 | 17.72 | 14.17 | 40.07 | 20.26 | 17.23 | 11.42 | 1.44 | 18.35 | 58.42 | 42.95 | 9.59 | 45.69 | 29.77 | 36.35 |
| | $\tau' = 0.07$ **(Ours)** | 45.80 | **42.32** | **23.03** | **73.15** | **18.26** | 20.52 | 44.02 | 24.54 | **18.14** | 2.26 | **2.61** | **20.15** | **63.73** | **48.66** | **12.60** | 29.51 | **30.58** | **39.09** |
| | $\tau' = 0.09$ | 38.58 | 41.00 | 21.23 | 71.83 | 16.93 | **21.60** | 40.67 | **25.08** | 17.87 | 1.58 | 2.43 | 18.98 | 63.36 | 47.54 | 11.37 | 23.08 | 28.94 | 37.99 |
| | $\tau' = 0.10$ | 37.64 | 40.75 | 19.94 | 70.90 | 16.06 | 20.57 | 40.77 | 24.15 | 17.18 | 1.24 | 2.28 | 18.04 | 61.53 | 46.28 | 10.96 | 17.82 | 27.88 | 37.04 |

## D.5. Generalization across vision–language models

We further examine whether UCAT is tied to the CLIP-B/32 backbone. We fine-tune SLIP-B16 (Mu et al., 2022), CLIP-B/16 (Radford et al., 2021), and CLIP-B/32 (Radford et al., 2021) on TinyImageNet using the same zero-shot adversarial robustness (ZSAR) configuration (2-step PGD, $\epsilon = 1/255$) in Table 9. This evaluates the architecture-agnostic nature of our Dirichlet alignment.

*Table 9.* **Zero-shot adversarial robustness on different vision–language models (VLMs).** All methods are fine-tuned on TinyImageNet following TeCoA (Yu et al., 2024), adversarial training uses 2-step PGD (Madry et al., 2017) with $\epsilon = 1/255$. *Average* is the mean across datasets. *H* is the harmonic mean between Clean and the corresponding robust score.

| Methods | | TinyImageNet | CIFAR-10 | CIFAR-100 | STL10 | SUN397 | Food101 | Oxfordpets | Flowers102 | DTD | EuroSAT | FGVC Aircraft | ImageNet | Caltech101 | Caltech256 | StanfordCars | PCAM | Average | H |
|---|---|---|---|---|---|---|---|---|---|---|---|---|---|---|---|---|---|---|---|
| CLIP-B/16 (Radford et al., 2021) | Clean | 60.86 | 89.49 | 66.25 | 98.03 | 61.05 | 88.51 | 88.66 | 70.97 | 43.03 | 45.83 | 24.57 | 0.07 | 86.07 | 84.99 | 58.29 | 52.86 | 63.72 | |
| | AutoAttack | 0.00 | 0.00 | 0.03 | 0.00 | 0.01 | 0.00 | 0.00 | 0.02 | 0.00 | 0.08 | 0.06 | 0.00 | 0.01 | 0.00 | 0.03 | 0.00 | 0.01 | 0.03 |
| + UCAT (Ours) | Clean | 77.52 | 82.02 | 57.34 | 92.66 | 43.78 | 54.76 | 73.75 | 49.86 | 30.16 | 24.25 | 12.39 | 0.06 | 82.21 | 73.72 | 37.51 | 54.50 | 52.91 | |
| | AutoAttack | 50.90 | 46.15 | 26.05 | 76.83 | 19.74 | 21.56 | 45.08 | 27.13 | 17.66 | 2.37 | 3.84 | 0.00 | 65.09 | 52.64 | 13.02 | 20.65 | 30.54 | 39.05 |
| SLIP-B/16 (Mu et al., 2022) | Clean | 36.74 | 78.97 | 44.22 | 94.28 | 52.71 | 59.70 | 31.67 | 59.67 | 21.65 | 19.87 | 5.79 | 38.61 | 75.15 | 62.71 | 5.85 | 48.96 | 46.03 | |
| | AutoAttack | 0.04 | 0.00 | 0.02 | 0.01 | 0.02 | 0.03 | 0.00 | 0.07 | 0.11 | 0.04 | 0.00 | 0.02 | 0.04 | 0.03 | 0.00 | 0.01 | 0.02 | 0.05 |
| + UCAT (Ours) | Clean | 51.90 | 70.14 | 38.51 | 86.76 | 41.86 | 26.59 | 25.29 | 38.19 | 16.22 | 13.50 | 3.75 | 26.42 | 70.12 | 51.07 | 3.37 | 50.29 | 38.37 | |
| | AutoAttack | 25.52 | 34.97 | 16.04 | 66.73 | 19.13 | 8.77 | 7.39 | 16.34 | 1.23 | 0.75 | | 11.58 | 49.03 | 29.35 | 0.57 | 30.92 | 20.40 | 26.68 |
| CLIP-B/32 (Radford et al., 2021) | Clean | 57.27 | 88.05 | 60.47 | 97.04 | 57.27 | 83.89 | 87.35 | 65.52 | 40.80 | 42.50 | 20.13 | 59.15 | 85.33 | 81.72 | 52.02 | 52.24 | 64.42 | |
| | AutoAttack | 1.26 | 6.47 | 0.33 | 30.70 | 0.70 | 9.73 | 4.80 | 1.11 | 0.11 | 0.10 | 0.00 | 1.00 | 19.11 | 13.56 | 0.37 | 0.00 | 5.58 | 10.28 |
| + UCAT (Ours) | Clean | 74.46 | 81.81 | 54.45 | 91.88 | 41.06 | 53.58 | 74.16 | 47.57 | 31.92 | 19.29 | 10.95 | 43.20 | 82.39 | 71.53 | 37.32 | 51.20 | 54.17 | |
| | AutoAttack | 45.80 | 42.32 | 23.03 | 73.15 | 18.26 | 20.52 | 44.02 | 24.54 | 18.14 | 2.26 | 2.61 | 20.15 | 63.73 | 48.66 | 12.60 | 29.51 | 30.58 | 39.09 |

## D.6. Comparison with classical adversarial training baselines

Finally, we examine whether classical adversarial training methods (AT) can serve as a baseline for zero-shot adversarial robustness (ZSAR). For fairness, all AT baselines (TRADES (Zhang et al., 2019b), ACAT (Addepalli et al., 2022), DKL (Cui et al., 2024)) are fine-tuned on TinyImageNet using the same 2-step PGD configuration ($\epsilon = 1/255$) with their *native objective, supervision signal, and optimization hyperparameters*. These models are then evaluated in a zero-shot manner on downstream ZSAR benchmarks, where no task-specific labels are used during testing. Although these AT methods were not originally developed for zero-shot scenarios, they still achieve strong performance under ZSAR (Table 10), demonstrating the general effectiveness of their theoretical formulations. However, compared with these supervised AT approaches, our UCAT method, which is explicitly designed to preserve open-set semantic alignment, achieves consistently better overall ZSAR performance, particularly in terms of the harmonic mean between clean and robust accuracy.

*Table 10.* **Comparison with representative adversarial training methods zero-shot adversarial robustness (ZSAR) setting.** All methods are fine-tuned on TinyImageNet following TeCoA (Yu et al., 2024), adversarial training uses 2-step PGD (Madry et al., 2017) with $\epsilon = 1/255$. *Average* is the mean across datasets. *H* is the harmonic mean between Clean and the corresponding robust score. Best and second-best are in **bold** and underline.

| | Methods | TinyImageNet | CIFAR-10 | CIFAR-100 | STL10 | SUN397 | Food101 | Oxfordpets | Flowers102 | DTD | EuroSAT | FGVC Aircraft | ImageNet | Caltech101 | Caltech256 | StanfordCars | PCAM | Average | H |
|---|---|---|---|---|---|---|---|---|---|---|---|---|---|---|---|---|---|---|---|
| Clean | CLIP (Radford et al., 2021) | 57.96 | 88.02 | 60.47 | 97.03 | 57.26 | 83.89 | 87.38 | 65.52 | 40.69 | 42.65 | 20.16 | 59.15 | 85.33 | 81.73 | 51.98 | 52.08 | 64.46 | |
| | TRADES (Zhang et al., 2019b) | 67.73 | 62.05 | 34.38 | 80.81 | 26.64 | 22.68 | 54.43 | 24.48 | 20.59 | 16.51 | 3.60 | 26.31 | 63.77 | 50.89 | 14.59 | 49.95 | 38.71 | |
| | ACAT (Addepalli et al., 2022) | 72.80 | 64.72 | 34.71 | 82.41 | 30.06 | 19.13 | 60.40 | 26.80 | 17.29 | 15.62 | 4.41 | 28.60 | 64.94 | 50.89 | 15.82 | 50.01 | 39.91 | |
| | DKL (Cui et al., 2024) | 70.84 | 65.31 | 35.61 | 82.54 | 30.11 | 21.11 | 55.11 | 25.94 | 21.17 | 16.26 | 3.96 | 27.18 | 65.84 | 52.00 | 14.20 | 48.97 | 39.76 | |
| | UCAT (Ours) | 74.46 | 81.81 | 54.45 | 91.88 | 41.06 | 53.58 | 74.16 | 47.57 | 31.92 | 19.29 | 10.95 | 43.20 | 82.39 | 71.53 | 37.32 | 51.20 | 54.17 | |
| AutoAttack | CLIP (Radford et al., 2021) | 1.26 | 6.47 | 0.33 | 30.70 | 0.70 | 9.73 | 4.80 | 1.11 | 0.11 | 0.10 | 0.00 | 1.00 | 19.11 | 13.56 | 0.37 | 0.00 | 5.58 | 10.28 |
| | TRADES (Zhang et al., 2019b) | 52.19 | 41.39 | 22.44 | 67.59 | 15.69 | 12.50 | 37.42 | 16.46 | 15.75 | 11.77 | 1.23 | 16.16 | 51.16 | 37.84 | 7.29 | 46.42 | 28.33 | 32.72 |
| | ACAT (Addepalli et al., 2022) | 51.85 | 34.54 | 16.85 | 64.84 | 14.99 | 8.95 | 36.71 | 16.56 | 10.96 | 11.41 | 1.59 | 15.69 | 48.70 | 34.84 | 7.62 | 49.89 | 26.62 | 31.94 |
| | DKL (Cui et al., 2024) | 54.71 | 41.77 | 22.29 | 67.84 | 17.32 | 10.73 | 36.06 | 16.23 | 15.69 | 11.72 | 1.65 | 16.66 | 51.85 | 38.33 | 6.67 | 43.48 | 28.31 | 33.07 |
| | UCAT (Ours) | 45.80 | 42.32 | 23.03 | 73.15 | 18.26 | 20.52 | 44.02 | 24.54 | 18.14 | 2.26 | 2.61 | 20.15 | 63.73 | 48.66 | 12.60 | 29.51 | 30.58 | 39.09 |

Overall, our method remains consistently strong across both extended settings, confirming its robustness under larger datasets, stronger adversarial attacks, hyperparameter variations, and different contrastive VLM architectures.

