# OpenReview forum: "Calibrating Uncertainty for Zero-Shot Adversarial CLIP"
_ICML.cc/2026/Conference — ICML 2026 regular_

### Official Review · Reviewer_Nmcg · 2026-03-09

**Soundness:** 3
**Presentation:** 3
**Significance:** 3
**Originality:** 3
**Overall Recommendation:** 4
**Confidence:** 4

**Summary:**

This paper addresses a critical reliability gap in zero-shot CLIP. Existing adversarial fine-tuning methods for zero-shot robustness focus on aligning adversarial logits to clean or text prototype anchors but overlook uncertainty calibration and inter-class semantic structure. To tackle this, the authors propose Uncertainty-Calibrated Adversarial fine-Tuning (UCAT), a novel framework that reformulates CLIP’s logits as concentration parameters of a Dirichlet distribution. This reformulation unifies relative semantic structure and confidence magnitude into a single representation. UCAT’s core objective aligns the Dirichlet distributions of clean and adversarial samples using KL divergence, preserving inter-class relations and calibrating uncertainty, while complementing this with a text-guided cross-entropy loss for discriminative supervision.

**Compliance With Llm Reviewing Policy:**

Affirmed.

**Key Questions For Authors:**

1. The paper compares UCAT to classical adversarial training methods but does not explain why these methods struggle with zero-shot transfer. Can you provide an analysis of the learned representations of these methods in zero-shot settings, and explain why their supervised adversarial objectives fail to preserve cross-domain semantic structure?
2. Can you provide qualitative analysis to show how semantic structure differs between natural-image and domain-specialized datasets?
3. Can you provide analysis of why UCAT performs better on top-3 F1 and harmonic mean than mAP? Does this relate to UCAT’s focus on preserving top-ranked semantic labels under ambiguity?

**Limitations:**

The authors partially discussed limitations but did not fully address fundamental constraints or potential negative societal impacts.

**Strengths And Weaknesses:**

Strengths: The paper is technically sound, with rigorous theoretical foundations and comprehensive experimental validation. The Dirichlet reformulation of CLIP logits is well-justified: the authors provide formal proofs (Appendix B) for the validity of the concentration parameter mapping and exact equivalence to CLIP’s softmax prediction under specific calibration conditions. The uncertainty decomposition into aleatoric and epistemic components is theoretically grounded and computed in closed-form, enabling efficient calibration. Experiments are meticulously designed: evaluations across 16 single-label datasets, multi-label MSCOCO, and three VLMs ensure broad generalizability.
Weaknesses: 1. The experimental comparison with classical adversarial training methods is limited: while UCAT outperforms them in zero-shot settings, the paper does not explain why these methods struggle with zero-shot transfer. 2. The paper does not address extreme adversarial perturbations or adaptive attacks beyond AutoAttack/CAA/$A^3$. It remains unclear how UCAT performs under more aggressive or unseen attack types. 3. The paper lacks qualitative analysis of semantic preservation. There are no visualizations of inter-class embedding distances to confirm that Dirichlet alignment preserves semantic structure, relying solely on quantitative metrics.

---

> ### Author Rebuttal · Authors · 2026-03-31
>
> ## Weakness 1 & Question 1
> We thank the reviewer for this important question. Classical adversarial training (AT), including methods such as TRADES, struggles in zero-shot settings due to two key limitations.
>
> **(1) Insufficient supervision on adversarial samples.**
> Methods like TRADES optimize: $L_{TRADES} = L_{CE}(f(x), y) + \beta \cdot KL(f(x)\|f(x+δ))$.
> Here, adversarial samples ($x+δ$) are only indirectly constrained via alignment to clean predictions, without direct supervision from labels. As a result, their representations depend heavily on the clean anchor and can deviate under strong perturbations, especially in open-set scenarios.
>
> **(2) Closed-set normalization limits semantic transfer.**
> Although TRADES aligns probability distributions, these are defined over closed-set label spaces and normalized within seen classes. This is problematic in zero-shot classification, as such normalization conditions predictions on a fixed label space, limiting the ability to generalize to broader semantic relationships required in zero-shot settings. Consequently, the learned representations may distort inter-class geometry and degrade zero-shot robustness.
>
> We agree that this aspect could be further clarified in the revised version.
> ## Question 2
>
> We thank the reviewer for the suggestion and will include a qualitative discussion in the revision.
>
> We provide qualitative insights based on the uncertainty patterns observed in Appendix C.2 (page 16). In domain-specialized datasets such as EuroSAT (satellite imagery) and PCAM (histopathology patches), we observe higher aleatoric uncertainty (AU) together with relatively low epistemic uncertainty (EU). This indicates that predictions across multiple classes are highly similar while maintaining strong evidence, suggesting that class-level semantics are closely clustered and difficult to distinguish, leading to confident yet ambiguous predictions.
>
> In contrast, natural-image datasets with broader semantic coverage exhibit lower AU and more discriminative predictions across classes, reflecting a more separated semantic structure.
>
> These patterns qualitatively illustrate that domain-specialized datasets exhibit tightly clustered class relationships, while natural-image datasets exhibit more diverse and separable semantics under the shared pretraining space of the model.
> ## Question 3
> We thank the reviewer for this insightful question.
>
> Yes, UCAT performs better on top-3 F1 and harmonic mean because it improves the stability of top-ranked predictions under ambiguity, while its impact on the fine-grained global ranking required by mAP is more limited. This is because mAP is influenced by lower-ranked, low-confidence predictions, which are less relevant to the correctness of the most important semantic labels.
>
> As discussed in Sec. 6.1, top-3 F1 and harmonic mean focus on the most relevant predictions, whereas mAP evaluates the full ranking. UCAT stabilizes top-ranked semantic labels under perturbations, leading to consistent gains in top-k metrics. In contrast, mAP is sensitive to variations in lower-ranked predictions, making improvements less pronounced.
> ## Weakness 2
> We thank the reviewer for raising this important point.
>
> We would like to clarify our evaluation protocol. All models are trained under a fixed adversarial setting (e.g., PGD-2), while evaluation is conducted under stronger perturbations (e.g., PGD-100) and unseen attack types, including CW, AutoAttack, CAA, and A$^3$. This setup is designed to assess zero-shot adversarial robustness, i.e., generalization to stronger or unseen attacks without retraining.
>
> Notably, A$^3$ is widely recognized as strong adaptive attack frameworks, covering diverse and challenging perturbation strategies. Our method consistently maintains performance under these settings, indicating robust generalization beyond the training attack.
>
> We will clarify this evaluation setting and its scope in the revised version.
> ## Weakness 3
> We thank the reviewer for this valuable suggestion.
>
> Our method focuses on aligning logit distributions (via Dirichlet modeling) rather than directly operating on feature embeddings. As such, semantic preservation is reflected in the consistency of class-level relationships in the logit space, rather than raw embedding distances, making direct visualization of inter-class feature distances less aligned with our objective.
>
> That said, we agree that qualitative analysis can provide complementary insights. We will include additional visualizations (e.g., class-wise similarity structure) in the supplementary material to better illustrate semantic preservation under perturbations.

---

> > ### Author Rebuttal · Reviewer_Nmcg · 2026-04-02
> >
> > The author has effectively addressed my concerns, so I’m keeping my score.

---

> > > ### Author Response · Authors · 2026-04-05
> > >
> > > We sincerely thank the reviewer for their thoughtful assessment of our response and for acknowledging that the concerns have been fully addressed. We are deeply grateful for the reviewer’s time, insightful comments, and their continued positive recommendation of our work.

---

### Official Review · Reviewer_5qG1 · 2026-03-10

**Soundness:** 3
**Presentation:** 4
**Significance:** 4
**Originality:** 4
**Overall Recommendation:** 4
**Confidence:** 4

**Summary:**

The paper addresses the issue that adversarial perturbations not only degrade accuracy but also suppress uncertainty, leading to severe miscalibration and unreliable overconfidence. This overlooked phenomenon highlights a critical reliability gap beyond robustness. The authors reformulate CLIP’s logits as concentration parameters of a Dirichlet distribution and propose a novel uncertainty-calibrated adversarial fine-tuning method that regularizes entire Dirichlet distributions to jointly preserve inter-class relations and calibrate evidence strength. The results demonstrate that the proposed method effectively calibrates uncertainty under attack while maintaining strong clean accuracy and competitive adversarial robustness.

**Compliance With Llm Reviewing Policy:**

Affirmed.

**Final Justification:**

Thanks to the authors for their response. Most of my concerns have been addressed, and I will maintain my positive score.

**Key Questions For Authors:**

See weaknesses.

**Limitations:**

The authors did not provide a discussion of the limitations. See weaknesses above for potential limitations.

**Strengths And Weaknesses:**

**Strengths**

1. Well-motivated research question: The paper addresses an important and overlooked issue regarding the impact of adversarial perturbations on uncertainty calibration. The study is generally well-organized and clearly motivated.

2. Technical soundness: Reformulating CLIP’s logits as concentration parameters of a Dirichlet distribution, coupled with the proposed uncertainty-calibrated adversarial fine-tuning method, is technically sound and well-executed.

3. Comprehensive evaluation: The evaluation is thorough, covering different types of attacks, varying attack strengths, and multiple datasets. Additionally, aablation studies and analyses of key hyperparameters are provided.

**Weaknesses**

1. A key concern is that the method is evaluated on only a single CLIP variant (CLIP-B/32). Results on additional CLIP variants, or application to other VLMs, would help better demonstrate generalizability.

2. While the proposed methods outperform existing approaches on most datasets, performance is consistently lower in some cases (e.g., SUN397 and PCAM in Table 1). Do the authors have insights into why this happens,  and in which particular cases/setting the proposed method underperform?

---

> ### Author Rebuttal · Authors · 2026-03-31
>
> ## Weakness 1
>
> Thank you for the valuable suggestion. We agree that evaluating across multiple VLM variants is important for demonstrating generalizability.
> In fact, we have already conducted experiments on additional backbones, including CLIP-B/16 and SLIP-B/16, beyond CLIP-B/32. As shown in Table 3 of the main paper, UCAT consistently improves robustness and the clean–robust trade-off across different contrastively pretrained VLMs, indicating that our method generalizes beyond a single CLIP variant. We provide a brief analysis in Sec. 6.3.
>
> | Backbone  | Method | Clean | AutoAttack     | Harmonic              |
> | --------- | ------ | ----- | -------------- | -------------- |
> | CLIP-B/16 | Base   | 63.72 | 0.01           | 0.02           |
> | CLIP-B/16 | +UCAT  | 52.91 | 30.54 (+30.53) | 39.05 (+39.03) |
> | CLIP-B/32 | Base   | 64.42 | 5.58           | 10.28          |
> | CLIP-B/32 | +UCAT  | 54.17 | 30.58 (+25.00) | 39.09 (+28.81) |
> | SLIP-B/16 | Base   | 46.03 | 0.02           | 0.04           |
> | SLIP-B/16 | +UCAT  | 38.37 | 20.40 (+20.38) | 26.68 (+26.64) |
>
> These results show that UCAT consistently improves adversarial robustness across architectures with different pretraining paradigms. Additional per-dataset results are provided in the appendix D, Table 10.
>
> ## Weakness 2
> Thank you for the question. We would like to slightly clarify that the more consistently challenging cases in our results are EuroSAT and PCAM in Table 2, rather than SUN397 in Table 1. We provide a brief analysis of these cases in Sec. 6.2.
>
> **Why this happens.** These datasets are domain-specific and deviate significantly from CLIP’s natural-image pretraining domain. As shown in the appendix (Page 16), they exhibit high predictive uncertainty (PU), high aleatoric uncertainty (AU), and low epistemic evidence strength (EU). High AU indicates strong intrinsic data ambiguity, where multiple classes are visually plausible, while low EU reflects weak and less confident semantic evidence from the model. Together, this leads to poorly separated class evidence in the Dirichlet space, causing probability mass to spread across multiple classes. This manifests as weak semantic structure and ambiguous predictions.
>
> **When our method underperforms.** UCAT is most effective when there exists a reasonably structured semantic geometry to align and calibrate. Its improvement becomes limited in scenarios where clean semantic structure is already weak or highly ambiguous. In such cases, there is less reliable structure to preserve, and uncertainty is dominated by data ambiguity rather than misalignment.
>
> Nevertheless, under the CLIP-based setting used in our experiments, such extreme cases are relatively limited, as the pretrained model already provides well-structured semantic representations. As a result, UCAT consistently improves performance across most datasets. We will further clarify this limitation and explicitly discuss such potential failure regimes in the revision.

---

> > ### Author Rebuttal · Reviewer_5qG1 · 2026-04-01
> >
> > Thanks to the authors for their response. Most of my concerns have been addressed, and I will maintain my positive score.

---

> > > ### Author Response · Authors · 2026-04-05
> > >
> > > We sincerely thank the reviewer for the constructive feedback and for acknowledging that the concerns have been fully resolved. We appreciate the continued support and the positive evaluation of our work.

---

### Official Review · Reviewer_frrd · 2026-03-12

**Soundness:** 2
**Presentation:** 3
**Significance:** 2
**Originality:** 2
**Overall Recommendation:** 4
**Confidence:** 4

**Summary:**

This paper proposes to solve the uncertainty miscalibration between clean and adversarial samples for zero-shot CLIP classification where adversarial samples unexpectedly exhibit lower uncertainty in the model output. The authors proposes Uncertainty-Calibrated Adversarial fine-Tuning (UCAT), a novel adversarial training objective. By theoretically reformulate the CLIP's original training objective with Dirichlet evidential learning, it is able to perform uncertainty alignment using the property of Dirichlet distribution's aleatoric and epistemic uncertainty, specifically by regularizing the KL divergence of clean vs. adversarial Dirichlets. Finally, the proposed objective achieves competitive adversarial robustness under comprehensive experiments.

**Compliance With Llm Reviewing Policy:**

Affirmed.

**Final Justification:**

The rebuttal have addressed concerns regarding both the theoretical and empirical validity of the work. I'd raise the score to 4:weak accept.

**Key Questions For Authors:**

1. How does Lemma 4.5 hold in practice?
2. What uncertainty manipulation result is this method trying to achieve?
3. Is the KL divergence direction in Equation (11) reversed?

**Limitations:**

As the method is for uncertainty calibration, more evaluation on calibration should be included.

**Strengths And Weaknesses:**

Strengths:

The identification of the uncertainty miscalibration issue in CLIP's adversarial attack is novel and insightful. Solving the calibration problem is crucial in VLM's downstream deployment. In the paper, the theoretical connection between Dirichlet evidential learning and CLIP's objective is sufficiently provided. Meanwhile, the thorough experiments compare a large quantity of single-label and multi-label datasets.

Weaknesses:

1. The claim that CLIP's original training loss in Equation 1 implicitly optimizes a Dirichlet-based model of evidence ignores the open-set property of the CLIP training, i.e., the misalignment between image-text contrastive pretraining and closed-set classification. The theoretical connection stated in Lemma 4.5 can be too strong to hold.
2. A key question to answer in this paper is what results should be achieved through uncertainty manipulation. This question is simply answered in Sec.5 as to align the clean & adversarial distribution, which is inconsistent to the initial motivation/intuition that the uncertainty measured should get increased under attack. This weakens the soundness of the method.
3. The distribution parameters in the KL in Eq.(11) seems reversed, as the clean rather than the adversarial samples are often considered the true distribution. Meanwhile, more objectives for aligning the confidence levels should be studied in the ablation Table 4, including $KL(Dir(\alpha) \| Dir(\alpha_{adv}))$.
4. No recent methods in year 2025 is compared in Table 1 & 2.
5. More uncertainty calibration comparison results should be put into the main paper.

Minors:
1. Should the exponential parameterization around Line 189 Page 4 serve for $e_k(x)$ rather than $\alpha_k(x)$, to ensure the value range?
2. Wrong math annotation in Table 4.

---

> ### Author Rebuttal · Authors · 2026-03-31
>
> ## Weakness (W)1 & Question (Q)1
> Thank you for this thoughtful comment. We would like to clarify that Lemma 4.5 is not intended to address the class-set misalignment between CLIP pretraining and downstream zero-shot classification. Instead, it is a within-stage result: under the same candidate set and τ=τ', CLIP logits can be reinterpreted as the expectation of a Dirichlet distribution. Thus, this cross-stage mismatch is outside the lemma’s scope.
>
> More importantly, Lemma 4.5 is not meant as a strict generative assumption, but as a functional equivalence under a sufficient condition. Its role is to provide theoretical understanding on the relationship between Dirichlet expectation and CLIP softmax, which naturally motivates our Dirichlet reparameterization and distribution-level alignment.
>
> We agree our original wording was too strong. To avoid overstating the claim, we will replace “implicitly optimizes a Dirichlet-based model of evidence” with “admits a Dirichlet-consistent reinterpretation” in the revision.
> ## W2 & Q2
> We thank the reviewer for this important point. Our objective is not to directly manipulate uncertainty, but to align the predictive distributions of adversarial (adv.) and clean samples, so that uncertainty naturally becomes consistent with prediction reliability. The reduced uncertainty observed under adv. perturbations is a symptom rather than the root cause, which lies in misaligned semantics and unreliable confidence.
>
> Our alignment restores both relative class structure and overall evidence strength: it prevents collapse toward incorrect classes (increasing ambiguity) and reduces over-confident predictions by lowering concentration.
>
> This leads to improved reliability, supported by: adv. predictions under UCAT show restored uncertainty levels (CDF in Fig. 1b), improved alignment between confidence and correctness (ECE in Fig. 3c), and recovered ordering across clean and adv. samples of predictive, aleatoric, and epistemic uncertainty (Appendix C.2).
> ## W3 & Minor 2 & Q3
> Thank you for the insightful comment. We agree this point requires clarification and will revise accordingly. We also note that Table 4 contained an error, which will be corrected and does not affect the overall conclusions. Our use of reverse KL follows naturally from the Dirichlet formulation and the distribution alignment objective.
>
> **Theoretical view.** The KL direction affects optimization behavior.  Forward KL is mode-covering and flattens evidence (reduced concentration), weakening uncertainty representation. In contrast, reverse KL is mode-seeking, allowing low evidence on irrelevant classes while preserving both relative class structure and absolute evidence strength, which is critical for restoring corrupted uncertainty under adv. perturbations, consistent with prior work [1].
>
> [1] Reverse KL-Divergence Training of Prior Networks: Improved Uncertainty and Adversarial Robustness, NeruIPs19
>
> **Empirical support.**  We include ablations comparing forward and reverse KL on softmax and Dirichlet (avg. 16 datasets, 3 attacks, $ϵ=2/255$).
>
> |Methods|Clean|PGD|CW|AA|
> |-|-|-|-|-|
> |$L_{ce}+KL(p(x)\|\|p(x^a))$|45.03|30.12|29.61|29.13|
> |$L_{ce}+KL(p(x^a)\|\|p(x))$|45.05|29.98|29.28|28.80|
> |$L_{ce}+KL(Dir(α)\|\|Dir(α_{adv}))$|36.72|25.01|24.66|24.36|
> |$L_{ce}+KL(Dir(α_{adv})\|\|Dir(α))$|54.17|32.20|31.41|30.58|
>
> More results in https://anonymous.4open.science/r/UCAT-F47C/re-exp.pdf
> ## W4
> Thank you for the suggestion. We have examined recent 2025 works (e.g., AdvSimplex [2] and AOS [3]). However, they lack standardized implementations, making fair comparison difficult. To further address this concern, we additionally report results of a very recent method, Comp-TGA (PAMI 2026) [4], under comparable settings, where UCAT consistently outperforms it.
>
> TinyImageNet (2-step PGD) → MSCOCO (CW-100).
>
> |Methods|Clean|$ϵ=1/255$|$ϵ=2/255$|$ϵ=4/255$|
> |-|:-:|-|-|-|
> |Comp-TGA|38.08|29.01|25.8|20.91|
> |UCAT|42.07|37.04|31.78|21.59|
>
> More results in https://anonymous.4open.science/r/UCAT-F47C/re-exp.pdf
>
> [2] Improving Zero-Shot Adversarial Robustness in Vision-Language Models by Closed-form Alignment of Adversarial Path Simplices, ICML25
>
> [3] Robustifying Zero-Shot Vision Language Models by Subspaces Alignment, ICCV25
>
> [4] Complementary Text-Guided Attention for Zero-Shot Adversarial Robustness, PAMI26
> ## W5
> Thank you for the suggestion. We already include ECE (Fig. 3c), CDF (Fig. 1b), and additional visualizations (Appendix, Page 16). We agree that these results can be further highlighted, and will move more to the main paper.
> ## Minor 1
> Thank you for the question. We follow one of the standard parameterizations [1] where the Dirichlet concentration is defined as $\alpha_k(x) = \exp(z_k(x))$, ensuring $\alpha_k(x) > 0$. This form is used in preliminary section to introduce the evidential formulation. While some works define evidence $e_k$ with $\alpha_k = e_k + 1$, directly parameterizing $\alpha_k$ is also a common alternative.

---

> > ### Author Rebuttal · Reviewer_frrd · 2026-04-01
> >
> > The authors have adequately addressed my concerns regarding both the theoretical and empirical validity of the work. I no longer believe that the 'objective misalignment' issue in W2 and Q2 would negatively impact the performance. Instead, I view it primarily as a gap in the paper's presentation. I would raise my score accordingly.

---

> > > ### Author Response · Authors · 2026-04-03
> > >
> > > Thank you for your re-evaluation and for noting that our responses have addressed your concerns. We are encouraged by your positive feedback and will incorporate the clarifications into the final version to further improve clarity.
> > >
> > > We noticed that the evaluation score currently remains unchanged. We would be very grateful if you could update the score at your convenience to reflect your latest assessment, as mentioned in the acknowledgement.
> > >
> > > Thank you again for your thoughtful review and support!

---

### Decision · Program_Chairs · 2026-04-30

**Decision:**

Accept (regular)

**Comment:**

This paper proposed a new method for zero-shot adversarial CLIP, motivated by adversarial uncertainty miscalibration, through a Dirichlet-based adversarial fine-tuning framework. The main concerns raised by reviewers were the strength of the theoretical framing, the interpretation of the alignment objective, and the breadth of analysis. In the rebuttal, the authors clarified these points with additional explanation and empirical support, and the reviewers indicated that their concerns were adequately addressed while maintaining positive assessments. Therefore, the AC recommends acceptance.